# Dual-satellite (Sentinel-2 and Landsat 8) remote sensing of supraglacial lakes in Greenland

Andrew G. Williamson[1], Alison F. Banwell[1,2], Ian C. Willis[1,2], Neil S. Arnold[1]

[1]Scott Polar Research Institute, University of Cambridge, Cambridge, UK

[2]Cooperative Institute for Research in Environmental Sciences, University of Colorado Boulder, Boulder, Colorado, USA

*Correspondence to:* Andrew G. Williamson (agw41@alumni.cam.ac.uk)

**Abstract.** Remote sensing is commonly used to monitor supraglacial lakes on the Greenland Ice Sheet; however, most satellite records must trade-off higher spatial resolution for higher temporal resolution (e.g. MODIS) or vice versa (e.g. Landsat). Here, we overcome this issue by developing and applying a dual-sensor method that can monitor changes to lake areas and volumes at high spatial resolution (10–30 m) with a frequent revisit time (~3 days). We achieve this by mosaicking imagery from the Landsat 8 OLI with imagery from the recently launched Sentinel-2 MSI for a ~12,000 km$^2$ area of West Greenland in the 2016 melt season. First, we validate a physically based method for calculating lake depths with Sentinel-2 by comparing measurements against those derived from the available contemporaneous Landsat 8 imagery; we find close correspondence between the two sets of values ($R^2 = 0.841$; RMSE $= 0.555$ m). This provides us with the methodological basis for automatically calculating lake areas, depths and volumes from all available Landsat 8 and Sentinel-2 images. These automatic methods are incorporated into an algorithm for Fully Automated Supraglacial lake Tracking at Enhanced Resolution (FASTER). The FASTER algorithm produces time series showing lake evolution during the 2016 melt season, including automated rapid ($\leq 4$ day) lake-drainage identification. With the dual Sentinel-2–Landsat 8 record, we identify 184 rapidly draining lakes, many more than identified with either imagery collection alone (93 with Sentinel-2; 66 with Landsat 8), due to their inferior temporal resolution, or would be possible with MODIS, due to its omission of small lakes $< 0.125$ km$^2$. Finally, we estimate the water volumes drained into the GrIS during rapid lake-drainage events and, by analysing downscaled regional climate-model (RACMO2.3p2) runoff data, the water quantity that enters the GrIS via the moulins opened by such events. We find that during the lake-drainage events alone, the water drained by small lakes ($< 0.125$ km$^2$) is only 5.1% of the total water volume drained by all lakes. However, considering the total water volume entering the GrIS after lake drainage, the moulins opened by small lakes deliver 61.5% of the total water volume delivered via the moulins opened by large and small lakes; this is because there are more small lakes, allowing more moulins to open, and because small lakes are found at lower elevations than large lakes, where runoff is higher. These findings suggest that small lakes should be included in future remote sensing and modelling work.

## 1 Introduction

In the summer, supraglacial lakes (hereafter "lakes") form within the ablation zone of the Greenland Ice Sheet (GrIS), influencing the GrIS's accelerating mass loss (van den Broeke *et al*., 2016) in two main ways. First, because the lakes have low albedo, they can directly affect the surface mass balance through enhancing ablation relative to the surrounding bare ice (Lüthje *et al.*, 2006; Tedesco *et al.*, 2012). Second, many lakes affect the dynamic component of the GrIS's mass balance when they drain either "slowly" or "rapidly" in the mid- to late melt season (e.g. Palmer *et al*., 2011; Joughin *et al*., 2013; Chu, 2014;

Nienow *et al.*, 2017). Slowly draining lakes typically overtop and incise supraglacial streams in days to weeks (Hoffman *et al.*, 2011; Tedesco *et al.*, 2013), while rapidly draining lakes drain by hydrofracture in hours to days (Das *et al.*, 2008; Doyle *et al.*, 2013; Tedesco *et al.*, 2013; Stevens *et al.*, 2015).

Rapid lake drainage plays an important role in the GrIS's negative mass balance because the large volumes of lake water can reach the subglacial drainage system, perturbing it from a steady state, lowering subglacial effective pressure, and enhancing basal sliding over hours to days (Shepherd *et al.*, 2009; Schoof, 2010; Bartholomew *et al.*, 2011a, 2011b, 2012; Hoffman *et al.*, 2011; Banwell *et al.*, 2013, 2016; Tedesco *et al.*, 2013; Andrews *et al.*, 2014), particularly if the GrIS is underlain by sediment (Bougamont *et al.*, 2014; Kulessa *et al.*, 2017; Doyle *et al.*, 2018; Hofstede *et al.*, 2018). Rapid lake-drainage events also have two longer-term effects. First, they open moulins, either directly within lake basins (Das *et al.*, 2008; Tedesco *et al.*, 2013) or in the far field if perturbations in stress exceed the tensile strength of ice (Hoffman *et al.*, 2018), sometimes leading to a cascading lake-drainage process (Christoffersen *et al.*, 2018). These moulins deliver the bulk of surface meltwater to the ice-sheet bed (Koziol *et al.*, 2017), explaining the observations of increased ice velocities over monthly to seasonal timescales within some sectors of the GrIS (Zwally *et al.*, 2002; Joughin *et al.*, 2008, 2013, 2016; Bartholomew *et al.*, 2010; Colgan *et al.*, 2011; Hoffman *et al.*, 2011; Palmer *et al.*, 2011; Banwell *et al.*, 2013, 2016; Cowton *et al.*, 2013; Sole *et al.*, 2013; Tedstone *et al.*, 2014; Koziol and Arnold, 2018). Second, the fractures generated during drainage allow surface meltwater to reach the subfreezing ice underneath, potentially increasing the ice-deformation rate over longer timescales (Phillips *et al.*, 2010, 2013; Lüthi *et al.*, 2015), although the magnitude of this effect is unclear (Poinar *et al.*, 2017). Alternatively, the water might promote enhanced subglacial conduit formation due to increased viscous heat dissipation (Mankoff and Tulaczyk, 2017). Although rapidly and slowly draining lakes are distinct, they can influence each other synoptically if, for example, the water within a stream overflowing from a slowly draining lake reaches the ice-sheet bed, thus causing basal uplift or sliding, and thereby increasing the propensity for rapid lake drainage nearby (Tedesco *et al.*, 2013; Stevens *et al.*, 2015).

While lake drainage is known to affect ice dynamics over short (hourly to weekly) timescales, greater uncertainty surrounds its longer-term (seasonal to decadal) dynamic impacts (Nienow *et al.*, 2017). This is because the subglacial drainage system in land-terminating regions may evolve to higher hydraulic efficiency, or water may leak into poorly connected regions of the bed, producing subsequent ice-velocity slowdowns either in the late summer, winter or longer term (van de Wal *et al.*, 2008, 2015; Bartholomew *et al.*, 2010; Hoffman *et al.*, 2011, 2016; Sundal *et al.*, 2011; Sole *et al.*, 2013; Tedstone *et al.*, 2015; de Fleurian *et al.*, 2016; Stevens *et al.*, 2016). Despite this observed slowdown for some of the GrIS's ice-marginal regions, greater uncertainty surrounds the impact of lake drainage on ice dynamics within interior regions of the ice sheet, since fieldwork and modelling suggest that increased summer velocities may not be offset by later ice-velocity decreases (Doyle *et al.*, 2014; de Fleurian *et al.*, 2016), and it is unclear whether hydrofracture can occur within these regions, due to the thicker ice and limited crevassing (Dow *et al.*, 2014; Poinar *et al.*, 2015). These unknowns inland add to the uncertainty in predicting future mass loss from the GrIS. There is a need, therefore, to study the seasonal filling and drainage of lakes on the GrIS, and to understand its spatial distribution and inter-annual variation, in order to inform the boundary conditions for GrIS hydrology and ice-dynamic models (Banwell *et al.*, 2012, 2016; Leeson *et al.*, 2012; Arnold *et al.*, 2014; Koziol *et al.*, 2017).

Remote sensing has helped to fulfil this goal (Hock *et al.*, 2017; Nienow *et al.*, 2017), although it usually involves trading-off either higher spatial resolution for lower temporal resolution, or vice versa. For example, the Landsat and ASTER satellites have been used to monitor lake evolution (Sneed and Hamilton, 2007; McMillan *et al.*, 2007; Georgiou *et al.*, 2009; Arnold *et al.*, 2014; Banwell *et al.*, 2014; Legleiter *et al.*, 2014; Moussavi *et al.*, 2016; Pope *et al.*, 2016; Chen *et al.*, 2017; Miles *et al.*, 2017; Gledhill and Williamson, 2018; Macdonald *et al.*, 2018). While this work involves analysing lakes at spatial resolutions of 30 or 15 m, respectively, the best temporal resolution that can be achieved using these satellites is ~4 days and is often much

longer due to the satellites' orbital geometry and/or site-specific cloud cover, which can significantly affect the observational record on the GrIS (Selmes *et al.*, 2011; Williamson *et al.*, 2017). This presents an issue for identifying rapid lake drainage with confidence since hydrofracture usually occurs in hours (Das *et al.*, 2008; Doyle *et al.*, 2013; Tedesco *et al.*, 2013). An alternative approach involves tracking lakes at high temporal (sub-daily) resolution but at lower spatial resolution (~250–500 m) using MODIS imagery (Box and Ski, 2007; Sundal *et al.*, 2009; Selmes *et al.*, 2011, 2013; Liang *et al.*, 2012; Johansson and Brown, 2013; Johansson *et al.*, 2013; Morriss *et al.*, 2013; Fitzpatrick *et al.*, 2014; Everett *et al.*, 2016; Williamson *et al.*, 2017, 2018). However, this lower spatial resolution means that lakes < 0.125 km$^2$ cannot be confidently resolved (Fitzpatrick *et al.*, 2014; Williamson *et al.*, 2017) and even lakes that exceed this size are often omitted from the satellite record (Leeson *et al.*, 2013; Williamson *et al.*, 2017).

Because of the problems associated with the frequency or spatial resolution of these satellite records, it has been suggested that greater insights into GrIS hydrology might be gained if the images from multiple satellites could be used simultaneously (Pope *et al.*, 2016). Miles *et al.* (2017) were the first to present such a record of lake observations in West Greenland, combining imagery from the Sentinel-1 Synthetic Aperture Radar (SAR) (hereafter "Sentinel-1") and Landsat 8 Operational Land Imager (OLI) (hereafter "Landsat 8") satellites, and developing a method for tracking lakes at high spatial (30 m) and temporal resolution (~3 days). Using Sentinel-1 imagery facilitated lake detection through clouds and in darkness, enabling, for example, lake freeze-over in the autumn to be studied. This approach permitted the identification of many more lake-drainage events than would have been possible if either set of imagery had been used individually, as well as the drainage of numerous small lakes that could not have been identified with MODIS imagery (Miles *et al.*, 2017). Monitoring all lakes, including the smaller ones, many of which may also drain rapidly by hydrofracture, is important since recent work shows that a key determinant on subglacial-drainage development is the density of surface-to-bed moulins opened by hydrofracture, rather than the hydrofracture events themselves (e.g. Banwell *et al.*, 2016; Koziol *et al.*, 2017). However, since Miles *et al.* (2017) used radar imagery, lake water volumes could not be calculated, restricting the type of information that could be obtained.

The Sentinel-2 Multispectral Instrument (MSI) comprises the Sentinel-2A (launched in 2016) and Sentinel-2B (launched in 2017) satellites, which have 290 km swath widths, a combined 5-day revisit time at the equator (with an even shorter revisit time at the poles), and 10 m spatial resolution in the optical bands; Sentinel-2 also has a 12-bit radiometric resolution, the same as Landsat 8, which improves on earlier satellite records with their 8-bit (or lower) dynamic range. Within glaciology, Sentinel-2 data have been used to, for example, map valley-glacier extents (Kääb *et al.*, 2016; Paul *et al.*, 2016), monitor changes to ice-dammed lakes (Kjeldsen *et al.*, 2017), and cross-compare ice-albedo products (Naegeli *et al.*, 2017); this research indicates that Sentinel-2 can be reliably combined with Landsat 8 since they produce similar results. Thus, Sentinel-2 imagery offers great potential for determining the changing volumes of lakes on the GrIS, for resolving smaller lakes, and for calculating volumes with higher accuracy than is possible with MODIS (Williamson *et al.*, 2017).

In this study, our objective is to present an automatic method for monitoring the evolution and drainage of lakes on the GrIS using a combination of Sentinel-2 and Landsat 8 imagery, which will allow the mosaicking of a high spatial resolution (10–30 m) record, with a frequent revisit time (approaching that of MODIS), something only possible by using the two sets of imagery simultaneously. The objective is addressed using four aims, which are to:

1. Trial new methods for calculating lake areas, depths and volumes from Sentinel-2 imagery and assess their accuracy against Landsat 8 for two days of overlapping imagery in 2016.
2. Apply the best methods for Sentinel-2 from (1), alongside existing methods for calculating lake areas, depths and volumes for Landsat 8, to all of the available 2016 melt season (May–October) imagery for a large study site (~12,000

km$^2$) in West Greenland. The aim is to apply these methods within an automated lake-tracking algorithm to produce time series of water-volume measurements for each lake in the study region to show their seasonal evolution.

3. Identify lakes that drain rapidly (in ≤ 4 days) using the automatic algorithm, separating these lakes into small (< 0.125 km$^2$) and large (≥ 0.125 km$^2$) categories, based on whether they could be identified with MODIS.

4. Quantify the runoff volumes routed into the GrIS both during the lake-drainage events themselves, and afterwards via moulins opened by hydrofracture, for the small and large lakes.

## 2 Data and methods

Here, we describe the study region (Sect. 2.1), the collection and pre-processing of the Landsat 8 and Sentinel-2 imagery (Sect. 2.2), the technique for delineating lake area (Sect. 2.3), the methods used to calculate lake depth and volume (Sect. 2.4), the approaches for automatically tracking lakes and identifying rapid lake drainage (Sect. 2.5), and the methods used to determine the runoff volumes that are routed into the GrIS's internal hydrological system following the opening of moulins by hydrofracture (Sect. 2.6).

### 2.1 Study region

Our analysis focuses on a ~12,000 km$^2$ area of West Greenland, extending ~110 km latitudinally and ~90 km from the ice margin, with this spatial extent chosen based on the full coverage of the original Sentinel-2 tiles (Fig. 1; Sect. 2.2.2). The region is primarily a land-terminating sector of the ice sheet, extending from the Sermeq Avannarleq outlet, which is just north of Jakobshavn Isbræ, near Ilulissat, to just south of Store Glacier in the Uummannaq district. We chose this study location because it is an area of high lake activity, having been the focus of many previous remote-sensing studies with which our results can be compared (e.g. Box and Ski, 2007; Selmes *et al.*, 2011; Fitzpatrick *et al.*, 2014; Miles *et al.*, 2017; Williamson *et al.*, 2017, 2018).

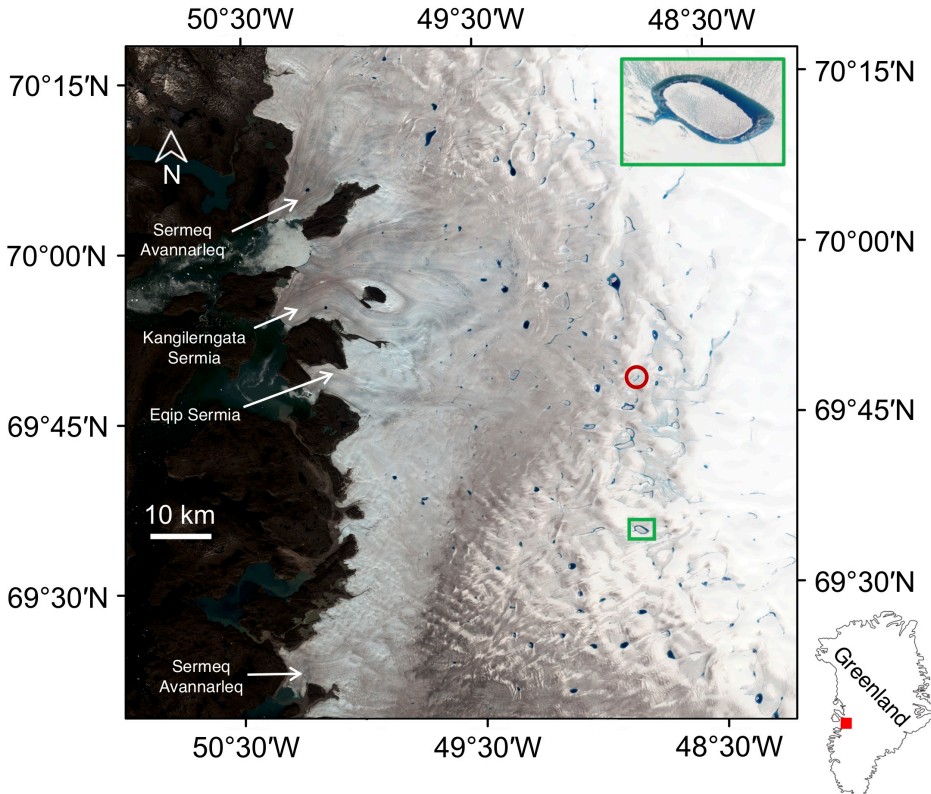

**Figure 1:** The ~12,000 km² study site within Greenland (inset). The background image is a Sentinel-2 RGB image from 11 July 2016 (see Table S1 for image details). The green box and enlarged subplot show a rapidly draining lake and the red circle shows a non-rapidly draining lake (cf. Fig. 5).

## 2.2 Satellite imagery collection and pre-processing

### 2.2.1 Landsat 8

17 Landsat 8 images from May to October 2016 (Table S2) were downloaded from the USGS Earth Explorer interface (http://earthexplorer.usgs.gov). These were level-1T, radiometrically and geometrically corrected images, which were distributed as raw digital numbers. We required the 30 m resolution data from bands 2 (blue; 0.452–0.512 μm), 3 (green; 0.533–0.590 μm), 4 (red; 0.636–0.673 μm) and 6 (shortwave infrared (SWIR); 1.566–1.651 μm), and the 15 m resolution data from band 8 (panchromatic; 0.503–0.676 μm). We used all available 2016 imagery that covered at least a portion of the study site, regardless of cloud cover. Since Landsat 8 images cover greater areas than Sentinel-2 images, we batch cropped the Landsat 8 images to the extent of the Sentinel-2 images using ArcGIS's 'Extract by Mask' tool. All of the tiles were reprojected (using bilinear resampling) to the WGS 84 UTM 22N geographic coordinate system (EPSG: 32622) for consistency with the Sentinel-2 images, and ice-marginal areas were removed with the Greenland Ice Mapping Project (GIMP) ice-sheet mask (Howat *et al.*, 2014). The raw digital numbers were converted to top-of-atmosphere (TOA) reflectance using the image metadata and the USGS Landsat 8 equations (available at: https://landsat.usgs.gov/landsat-8-l8-data-users-handbook-section-5). Landsat 8 TOA values adequately represent surface reflectance in Greenland (Pope *et al.*, 2016) and have been used previously for studying GrIS hydrology (Pope *et al.*, 2016; Miles *et al.*, 2017; Williamson *et al.*, 2017, 2018; Macdonald *et al.*, 2018). Our cloud-masking procedure involved marking pixels as cloudy when their band-6 (SWIR) TOA reflectance value exceeded 0.100 (Fig. 2), a method used for MODIS imagery albeit requiring a higher threshold value of 0.150 (Williamson *et al.*, 2017). We chose this lower threshold based on manual inspection of the pixels marked as cloudy against clouds visible on the original images. To reduce any uncertainty in the cloud-filtering technique, we then dilated the cloud mask by 200 m (just

over six Landsat 8 pixels), so that we could be confident that all clouds and their nearby shadows had been marked as 'no data'
and would not affect the subsequent analyses. The images were manually checked for shadowing elsewhere, with any shadows
filtered where present.

### 2.2.2 Sentinel-2

39 Sentinel-2A level-1C images from May to October 2016 (Table S1) were downloaded from the Amazon S3 Sentinel-2
database (http://sentinel-s2-l1c.s3-website.eu-central-1.amazonaws.com). The Sentinel-2 data were distributed as TOA
reflectance values that were radiometrically and geometrically corrected, including ortho-rectification and spatial registration
to a global reference system with sub-pixel accuracy. We note that Sentinel-2 and Landsat 8 images are ortho-rectified using
different DEMs, which may produce slight offsets in lake locations (Kääb *et al.*, 2016; Paul *et al.*, 2016); two contemporaneous
image pairs (1 July and 31 July 2016) were therefore manually checked prior to analysis, but no obvious offset was observed.
We included all Sentinel-2 images from 2016 that had $\geq 20\%$ data cover of the study region and $\leq 75\%$ cloud cover. This
resulted in using 39 images from the 77 in total available in 2016, reducing the average temporal resolution from 2.0 to 3.9
days. We downloaded data from Sentinel-2's 10 m resolution bands 2 (blue; 0.460–0.520 μm), 3 (green; 0.534–0.582 μm) and
4 (red; 0.655–0.684 μm), and 20 m resolution data from band 11 (SWIR; 1.570–1.660 μm). Ice-marginal areas were removed
using the GIMP ice-sheet mask (Howat *et al.*, 2014). We used a cloud-masking procedure similar to that for Landsat 8, where
pixels were assumed to be clouds and were marked as 'no data' when the TOA value exceeded a threshold of 0.140 in band
11 (SWIR), after the band-11 data had been interpolated (using nearest-neighbour resampling) to 10 m resolution for
consistency with the optical bands (Fig. 2). This threshold was chosen by manually comparing the pixels identified as clouds
against background RGB images. As with the Landsat 8 images, we dilated the cloud mask by 200 m (10 Sentinel-2 pixels) to
account for any uncertainty in the cloud-masking procedure, and again the images were manually checked for shadowing
elsewhere, with any shadows filtered where present.

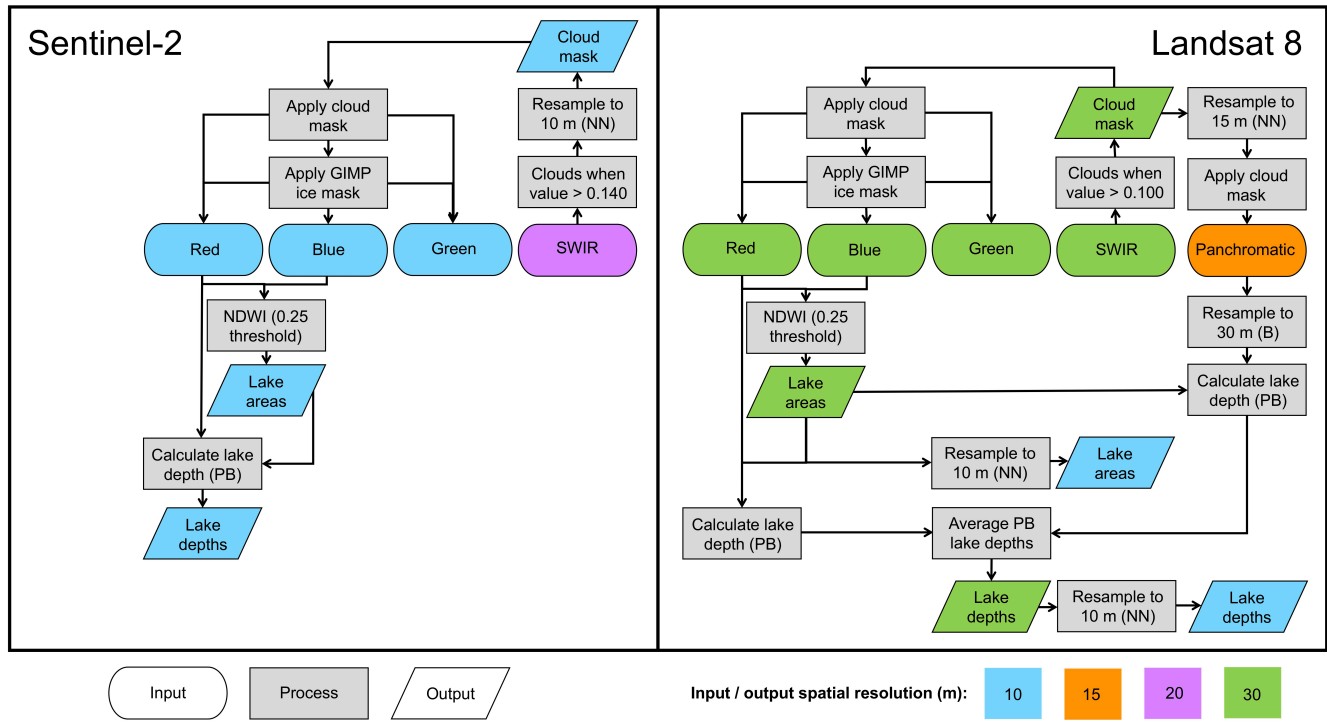

**Figure 2:** Summary of the methods applied to the Sentinel-2 and Landsat 8 input data to calculate lake areas using the NDWI and depths using the physically based ("PB" in this figure) method. In this figure, the interpolation techniques used are indicated by "NN" for nearest neighbour or "B" for bilinear. The methods applied to Landsat 8 are shown only once the images had been reprojected and batch cropped to the same extent as the Sentinel-2 images (Sect. 2.2.1). The lake-area outputs are compared between the two datasets as described in Sect. 2.3, and the physically based lake-depth outputs are compared as outlined in Sect. 2.4. When the empirical lake depth method for calculating Sentinel-2 lake depths was also evaluated, the final Landsat 8 depths at 10 m resolution were directly compared against the original Sentinel-2 input band data (at native 10 m resolution) within the lake outlines defined with the NDWI.

### 2.3    Lake-area delineation

Figure 2 summarises the overall method used to calculate lake areas and depths for the Landsat 8 and Sentinel-2 imagery, including the cloud-masking procedure described above, and the resampling required because the data were distributed at different spatial resolutions. We derived lake areas for the two sets of imagery using the Normalised Difference Water Index (NDWI) approach, which has been widely used previously for medium- to high-resolution imagery of the GrIS (e.g. Moussavi *et al.*, 2016; Miles *et al.*, 2017). There were two stages involved here. First, we applied various NDWI thresholds to the Sentinel-2 and Landsat 8 images, and compared the delineated lake boundaries against the lake perimeters in the background RGB images. We then qualitatively selected the NDWI threshold for each type of imagery based on the threshold that produced the closest match between the two. Based on this qualitative analysis, we chose NDWI thresholds of 0.25 for both types of imagery (Fig. 2). By varying the thresholds from these values to 0.251 and 0.249, respectively, the total lake area calculated across the whole image only changed by < 2%. The second stage involved comparing the areas of 594 lakes defined using the NDWI for the contemporaneous Landsat 8 and Sentinel-2 images from 1 July (collected in < 90 minutes of each other) and 31 July (collected in < 45 minutes of each other). This gave an extremely close agreement between the two sets of lake areas ($R^2$ = 0.999; RMSE = 0.007 km$^2$, equivalent to seven Sentinel-2 pixels) without any bias, so we were confident that the NDWI approach applied to the two types of imagery reproduced the same lake areas (Fig. S1). Using these NDWI thresholds, we created binary (i.e. lake and non-lake) masks for each day of imagery for the two satellites. Since the Landsat 8 optical-band data were at 30 m native resolution, we resampled them to 10 m resolution (using nearest-neighbour resampling) for

consistency with the resolution of the Sentinel-2 data (Fig. 2). From the binary images, we removed groups of < 5 pixels in total and linear features < 2 pixels wide, since these were likely to represent areas of mixed slush or supraglacial streams, as opposed to lakes (Pope, 2016; Pope et al., 2016).

## 2.4    Lake depth and volume estimates

### 2.4.1    Landsat 8

For each Landsat 8 image, we calculated the lake depths and volumes using the physically based method of Pope (2016) and Pope et al. (2016), based on Sneed and Hamilton's (2007) original method for ASTER imagery. This approach is based on the premise that there is a measurable change in the reflectance of a pixel within a lake according to its depth, since deeper water causes higher attenuation of the optical wavelengths within the water column. Lake depth ($z$) can therefore be calculated based on the satellite-measured reflectance for a pixel of interest ($R_{pix}$) and other lake properties:

$$z = \frac{[\ln \, (A_d - R_\infty) - \ln \, (R_{pix} - R_\infty)]}{g},$$    (1)

where $A_d$ is the lake-bottom albedo, $R_\infty$ is the reflectance for optically deep (> 40 m) water, and $g$ is the coefficient for the losses in upward and downward travel through a water column. For Landsat 8, we followed Pope et al.'s (2016) recommendation, taking an average of the depths calculated using the red and panchromatic band TOA reflectance data within the boundaries of the lakes (before they had been resampled to 10 m resolution for comparing the Sentinel-2 and Landsat 8 lake areas; Sect. 2.3) defined by the method described in Sect. 2.3. Since the panchromatic-band data were at 15 m resolution, we resampled them using bilinear interpolation to match the 30 m red-band resolution (Fig. 2). $A_d$ was calculated as the average reflectance in the relevant band for the ring of pixels immediately surrounding a lake, $R_\infty$ was determined from optically deep water in proglacial fjords on a scene-by-scene basis for each band, and we used $g$ values for the relevant Landsat 8 bands from Pope et al. (2016). Lake volume was calculated as the sum of lake depths, multiplied by the pixel area, within the lake outlines. We treated these Landsat 8 depths and volumes as ground-truth data as in Williamson et al. (2017).

### 2.4.2    Sentinel-2

Since no existing work has derived lake depths using Sentinel-2, we needed to formulate a new method. For this purpose, we used the Landsat 8 lake depths as our validation dataset. We conducted the validation on the two dates (1 July and 31 July 2016) with contemporaneous Landsat 8 and Sentinel-2 images (as described in Sect. 2.3). We chose to test both physically based and empirically based techniques to derive Sentinel-2 lake depths, noting at the outset that physical techniques are generally thought to be preferable over empirical ones since they do not require site- or time-specific tuning.

For the physically based technique, we tested whether the same method as applied to Landsat 8 (Eq. (1)) could be used on the Sentinel-2 TOA reflectance data. However, since Sentinel-2 does not collect panchromatic band measurements, we could only use individual Sentinel-2 bands to calculate lake depths. We applied this physically based technique to the red- and green-band data within the lake outlines defined with the NDWI (Sect. 2.3). We derived the value for $R_\infty$ as described above for Landsat 8. Since the lake-depth calculations are particularly sensitive to the $A_d$ value (Pope et al., 2016), it was critical to ensure that the lake-bottom albedo was correctly identified. Thus, to define  $A_d$, we dilated the lake by a ring of two pixels, and not one, to ensure that shallow water was not included due to the finer pixel resolution and due to any errors in the lake outlines derived using the NDWI. We also calculated new $g$ values for Sentinel-2's red and green bands using Pope et al.'s (2016) methods (Sect. S1).

Our empirically based approach involved deriving various lake depth-reflectance regression relationships, to determine which explained most variance in the data. We used the Landsat 8 lake depth data (dependent variable) and the Sentinel-2 TOA reflectance data for the three optical bands (independent variables) for each pixel within the lake outlines predicted in both sets of imagery, to determine which band and relationship produced the best match between the two datasets. To compare these values, we first resampled (using nearest-neighbour interpolation) the Landsat 8 depth data from 30 m to 10 m to match the resolution of the Sentinel-2 TOA reflectance data (Fig. 2).

To evaluate the performance of the empirical versus physical techniques, we calculated goodness-of-fit indicators for the Sentinel-2 and Landsat 8 measurements derived from the empirically based technique (applied to all optical bands) and physically based method (applied to the red and green bands) for the two validation dates (1 July and 31 July 2016) when contemporaneous Landsat 8 and Sentinel-2 images were available.

As for Landsat 8, Sentinel-2 lake volumes were calculated as the sum of the individual lake depths, multiplied by the pixel areas, within the lake boundaries.

## 2.5 Lake evolution and rapid lake-drainage identification

### 2.5.1 Time series of lake water volumes

Once validated, the new techniques to calculate lake areas, depths and volumes from Sentinel-2, as well as the existing methods for Landsat 8 (Sect 2.4), were applied to the satellite imagery within the Fully Automated Supraglacial lake Tracking at Enhanced Resolution (FASTER) algorithm to produce cloud- and ice-marginal-free 10 m resolution lake area and depth arrays for each day of the 2016 melt season for which either a Landsat 8 or Sentinel-2 image was available (Fig. 2). For the days (1 July and 31 July) when both Landsat 8 and Sentinel-2 imagery was available (as used for the comparisons above), in the FASTER algorithm, we used only the higher-resolution Sentinel-2 images. The FASTER algorithm is an adapted version of the Fully Automated Supraglacial lake Tracking (FAST) algorithm (Williamson *et al.*, 2017), which was developed for MODIS imagery. The FASTER algorithm involves creating an array mask to show the maximum extent of lakes within the region in the 2016 melt season, by superimposing the lake areas from each image. Within this maximum lake-extent mask, changes to lake areas and volumes were tracked between each consecutive image pair, with any lakes that were obscured (even partially) by cloud marked as 'no data'. We only tracked lakes that grew to $\geq 495$ pixels (i.e. 0.0495 km$^2$) at least once in the season, which is identical to the minimum threshold used by Miles *et al.* (2017), and is based on the minimum estimated lake size (approximated as a circle) required to force a fracture to the ice-sheet bed (Krawczynski *et al.*, 2009). It is encouraging that this minimum threshold size for lake tracking was over seven times larger than the error (0.007 km$^2$) associated with calculating lake area (Sect. 2.3; Fig. S1). While a lower tracking threshold could have been used, it would have significantly increased computational time and power required, alongside adding uncertainty to whether the tracked groups of pixels actually represented lakes. This tracking procedure produced time series for all lakes to show their evolution over the whole 2016 melt season.

### 2.5.2 Rapid lake-drainage identification

From the time series, a lake was classified as draining rapidly if two criteria were met: (i) it lost > 80% of its maximum seasonal volume in $\leq 4$ days (following Doyle *et al.*, 2013; Fitzpatrick *et al.*, 2014; Miles *et al.*, 2017; Williamson *et al.*, 2017, 2018); and (ii) it did not then refill on the subsequent day of cloud-free imagery by > 20% of the total water volume lost during the previous time period (following Miles *et al.*, 2017); the aim here was to filter false positives from the record. However, we also tested the sensitivity of the rapid lake-drainage identification methodology by varying the threshold by ± 10% (i.e. 70–

90%) for the critical-volume-loss threshold, ± 10% (i.e. 10–30%) for the critical-refilling threshold, and ± 1 day (i.e. 3–5 days) for the critical-timing threshold.

To determine how much extra information could be obtained from the finer spatial resolution satellite record, we compared the number of rapidly draining lakes identified that grew to $\geq 0.125$ km$^2$ (which would be resolvable by MODIS) with the number that never grew to this size (which would not be resolvable by MODIS) at least once in the season. We defined the drainage date as the midpoint between the date of drainage initiation and cessation, and identified the precision of the drainage date as half of this value. We conducted three sets of analyses: one for each set of imagery individually, and a third for both sets together; the intention here was to quantify how mosaicking the dual-satellite record improved the identification of lake-drainage events compared with using either record alone. The water volumes reaching the GrIS's internal hydrological system from the small and large lakes during the drainage events themselves were determined using the lake-volume measurements on the day of drainage.

## 2.6    Runoff deliveries following moulin opening

Using the dual Sentinel-2 and Landsat 8 record, the locations and timings of moulin openings by 'large' and 'small' rapidly draining lakes were identified. Then, at these moulin locations, the runoff volumes that subsequently entered the ice sheet were determined using statistically downscaled daily 1 km resolution RACMO2.3p2 runoff data (Noël *et al.*, 2018). Here, "runoff" was defined as melt plus rainfall minus any refreezing in snow (Noël *et al.*, 2018). These data were reprojected from Polar Stereographic (EPSG: 3413) to WGS 84 UTM zone 22N (EPSG: 32622) for consistency with the other data, and resampled to 100 m resolution using bilinear resampling. Then, the ice-surface catchment for each rapidly draining lake was delineated using MATLAB's 'watershed' function, applied to the GIMP ice-surface-elevation data (Howat *et al.*, 2014). The elevation data were first coarsened using bilinear resampling to 100 m resolution from 30 m native resolution. For each of the days after rapid lake drainage had finished, it was assumed that all of the runoff within a lake's catchment reached the moulin in that catchment instantaneously (i.e. no flow-delay algorithm was applied) and entered the GrIS. This method therefore assumes that once a moulin has opened at a lake-drainage site, it remains open for the remainder of the melt season. This allowed first-order comparisons between cumulative runoff routed into the GrIS via the moulins opened by small and large lake-drainage events.

## 3    Results

### 3.1    Sentinel-2 lake-depth estimates

Table 1 shows the results of the lake-depth calculations using the physically and empirically based techniques applied to imagery from 1 July and 31 July 2016 when contemporaneous Landsat 8 and Sentinel-2 images were available. The physically based method applied to the red and green bands (Figs. 3 and S2, respectively) performed slightly worse (for the red band: $R^2$ = 0.841 and RMSE = 0.555 m; for the green band: $R^2$ = 0.876 and RMSE = 0.488 m) than the best empirical method (Fig. 4) when a power-law regression was applied to the data ($R^2$ = 0.889 and RMSE = 0.447 m). Figures S3 and S4 respectively show the data for the empirical technique applied to the Sentinel-2 TOA reflectance and Landsat 8 lake depths for the worse-performing Sentinel-2 green and blue bands (Table 1). The physically based method applied to the red-band data performed better on 1 July, where the relationship between Sentinel-2-derived depths and Landsat 8 depths was more linear (Fig. 3, blue markers) than on 31 July (Fig. 3, red markers), where the relationship was more curvilinear. This is because the depths calculated with Sentinel-2 on 31 July were limited to ~3.5 m, while higher depths (> 4 m) were reported on 1 July (Fig. 3). Although less distinct, the best empirical relationship also differed slightly in performance between the two dates (Fig. 4). Section 4.1 discusses the possible reasons for the under-measurement of lake depths with Sentinel-2 on 31 July compared with

1 July. The physically based method applied to the green-band data performed similarly on both validation dates (Fig. S2).

Although application of the physically based technique to the green band produced a slightly higher $R^2$ and lower RMSE compared with the physically based method applied to the red-band data, the depths estimated with Sentinel-2 were unrealistically high compared with those from Landsat 8: Sentinel-2 reports a maximum depth of ~19 m, comparing with an equivalent value of ~5.5 m for Landsat 8 (Fig. S2). This produced more scatter for the green band than the red band physical method (Table 1).

Although the physically based method performed slightly worse than the empirical techniques, the physical method is preferable because it can be applied across wide areas of the GrIS and in different years without site- or time-specific tuning; it is likely that a different empirical relationship would have better represented the data for a different area of the GrIS or in a different year. We therefore carried forward the physically based method applied to the red band into the lake-tracking approach. We selected the red band instead of the green band because of the large difference between the depths calculated

with the two satellites at higher values when using the green band. We defined the error on all of the subsequently calculated lake-depth (and therefore lake-volume) measurements for Sentinel-2 using the RMSE of 0.555 m, and treated the Landsat 8 measurements as ground-truth data, meaning they did not have errors associated with them.

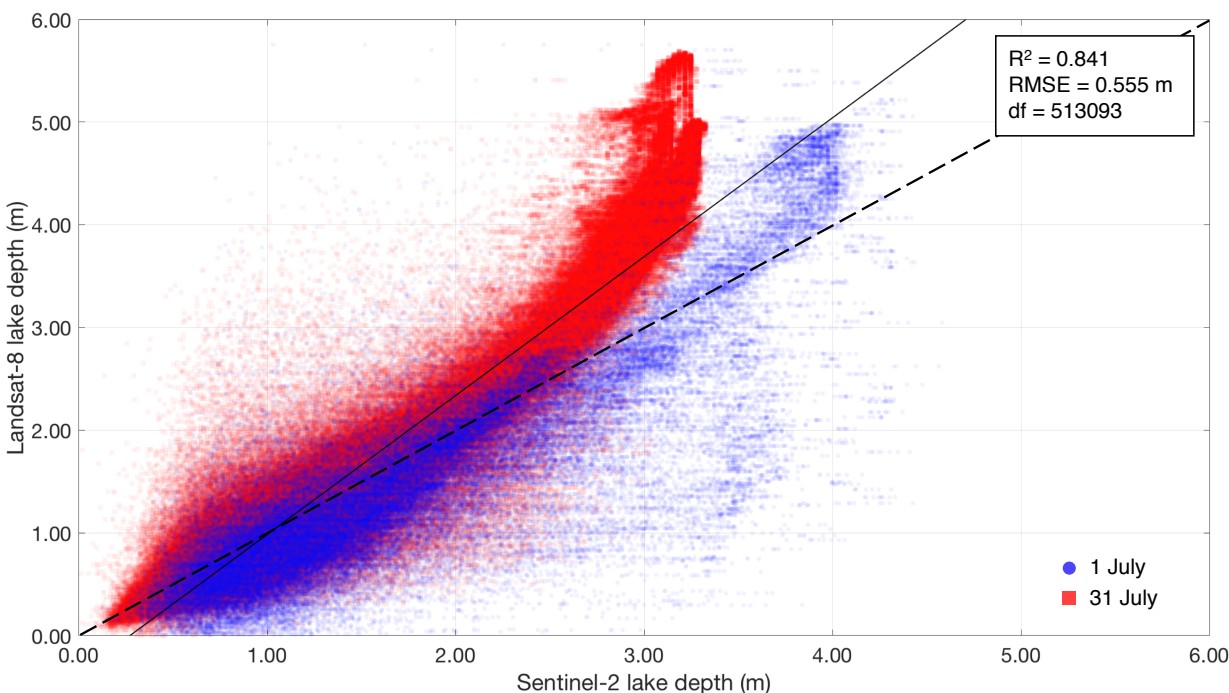

**Figure 3:** Comparison of lake depths calculated using the physically based method for Sentinel-2 (with the red band) and for

Landsat 8 (with the average depths from the red and panchromatic bands). Degrees of freedom ("df" in this figure) = 513,093. The solid black line shows an ordinary least-squares (OLS) linear regression and the dashed black line shows a 1:1 relation. The $R^2$ value indicates that the regression explains 84.1% of the variance in the data. The RMSE of 0.555 m shows the error associated with calculating the Sentinel-2 lake depths using this relationship.

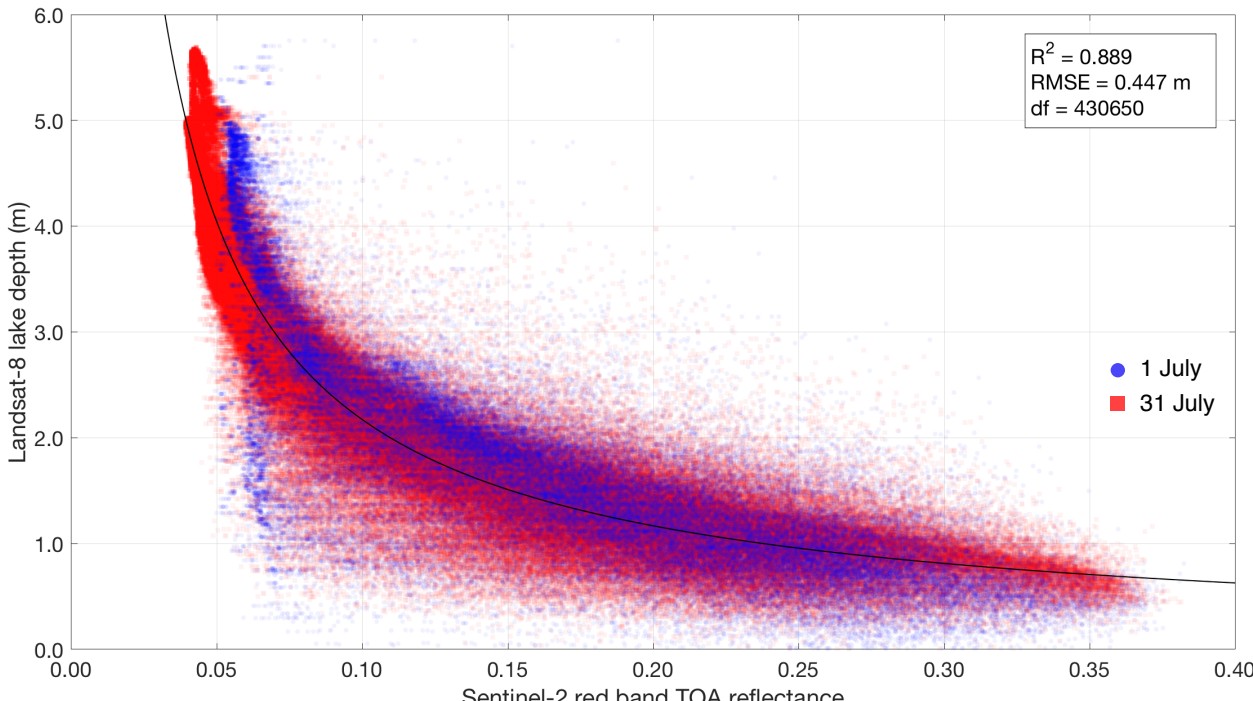

**Figure 4:** The empirical power law regression (solid black curve, equation $y = 0.2764x^{-0.8952}$) between Sentinel-2 red band TOA reflectance and Landsat 8 lake depth. Degrees of freedom ("df" in this figure) = 430,650. The $R^2$ value indicates that the regression explains 88.9% of the variance in the data. The RMSE of 0.447 m shows the error associated with calculating the Sentinel-2 lake depths using this relationship.

**Table 1:** Goodness-of-fit indicators for the empirical and physical techniques tested in this study for deriving Sentinel-2 lake depths, with validation against the Landsat 8 lake depth measurements, on 1 July and 31 July 2016. $R^2$ is the coefficient of determination, RMSE is the root mean square error, and SSE is the sum of squares due to error. The best performing (red band) regression relationship (i.e. the one with highest $R^2$, and lowest RMSE and SSE) among the empirical techniques is shown in bold italicised text. Data for the physical relation applied to Sentinel-2's green band are presented in Fig. S2, and data for the empirical relation applied to Sentinel-2's green and blue bands are presented in Figs. S3 and S4, respectively.

| Sentinel-2 band (technique) | Goodness-of-fit indicator | OLS regression | Power-law regression | Exponential regression |
|---|---|---|---|---|
| Red (empirical) | $R^2$ | 0.702 | ***0.889*** | 0.842 |
| | RMSE | 0.734 | ***0.448*** | 0.534 |
| | SSE (m$^3$) | $2.39 \times 10^5$ | ***$8.62 \times 10^4$*** | $1.23 \times 10^5$ |
| Green (empirical) | $R^2$ | 0.782 | 0.768 | 0.829 |
| | RMSE | 0.627 | 0.647 | 0.556 |
| | SSE (m$^3$) | $1.69 \times 10^5$ | $1.80 \times 10^5$ | $1.33 \times 10^5$ |
| Blue (empirical) | $R^2$ | 0.647 | 0.622 | 0.673 |
| | RMSE | 0.799 | 0.826 | 0.768 |
| | SSE (m$^3$) | $2.75 \times 10^5$ | $2.94 \times 10^5$ | $2.54 \times 10^5$ |
| Red (physical) | $R^2$ | 0.841 | | |
| | RMSE | 0.555 | – | – |
| | SSE (m$^3$) | $1.58 \times 10^5$ | | |
| Green (physical) | $R^2$ | 0.876 | | |
| | RMSE | 0.488 | – | – |
| | SSE (m$^3$) | $1.22 \times 10^5$ | | |

## 3.2    Lake evolution

Having verified the reliability of the lake area and depth techniques for both Sentinel-2 and Landsat 8, the automatic calculation methods were included in the FASTER algorithm to derive seasonal changes to lake areas and depths, and therefore volumes. The FASTER algorithm was applied to the Landsat 8 and Sentinel-2 image batches individually, as well as to both sets when combined into a dual record. Using the dual-satellite image collection produced an improvement to the temporal resolution of the dataset over the melt season (1 May to 30 September) from averages of 9.0 days (for Landsat 8) and 3.9 days (for Sentinel-2) to 2.8 days (for the dual record). The months of June and July had the most imagery available (both with 14 images) within the dual-satellite analysis. For the Landsat 8 individual analysis, the algorithm tracked changes to 453 lakes that grew to $\geq$ 0.0495 $km^2$ once in the season; equivalent numbers were 599 lakes for the Sentinel-2 analysis, and 690 lakes for the dual-satellite analysis. Using the dual record therefore involved tracking an additional 237 (or 91) lakes over the season than was possible with Landsat 8 (or Sentinel-2) alone.

The largest lake size varied between the analyses: 4.0 $km^2$ for Landsat 8 (recorded on 16 July 2016), and 8.6 $km^2$ for Sentinel-2 (recorded on 21 July 2016), which may be because there were no Landsat 8 images close to 21 July 2016. The maximum lake volumes recorded also varied between the two platforms: $1.1 \times 10^7$ $m^3$ for Landsat 8 (recorded on 15 July 2016) and $1.2 \times 10^7$ $m^3$ for Sentinel-2 (recorded on 14 July 2016). The mean lake size across all of the images from the dual Sentinel-2–Landsat 8 record was 0.137 $km^2$ (25[th] and 75[th] percentiles = 0.0075 $km^2$ and 0.129 $km^2$, respectively). This value is therefore just above (by 0.012 $km^2$) the threshold reporting size of MODIS, assuming two 250 m MODIS pixels are required to confidently classify lakes (Fitzpatrick *et al.*, 2014; Williamson *et al.*, 2017). Unpaired Student's *t*-tests between Sentinel-2 and Landsat 8 lake areas and volumes (from all of the imagery) confirmed that they were not significantly different with > 99% confidence ($t = 6.5$, degrees of freedom = 9503 for areas; $t = 11.4$, degrees of freedom = 6859 for volumes), justifying using the two imagery types together despite their resolution difference (10 m versus 30 m).

Using the full Sentinel-2–Landsat 8 dataset, the FASTER algorithm produced time series that documented changes to individual lake volumes over the season, samples of which are shown in Fig. 5. Total areal and volumetric changes across the whole region were calculated by summing the values for all lakes in the region. However, we found that cloud cover (which was masked from the images) often affected the observational record, and there were time periods, such as early July and the end of August, with a lot of missing data (Fig. 6). Figure 7 was therefore produced to normalise total lake areas and volumes against the proportion of the region visible, and this shows the estimated pattern of lake evolution on the GrIS: there was virtually no water in lakes before June, steady increases in total lake area and volume until the middle of July, and then a gradual decrease in total lake area and volume through the remainder of the season, with most lakes emptying by early September (Figs. 6 and 7). Dates with seemingly low total lake areas and volumes were usually explained by the low portion of the whole region visible in those images (Figs. 6 and 7). Finally, as in previous studies (e.g. Box and Ski, 2007; Georgiou *et al.*, 2009; Williamson *et al.*, 2017), we found a close correspondence between lake areas and volumes: comparing lake area and volume values from all dates produced an $R^2$ value of 0.73 ($p = 1.03 \times 10^{-10}$).

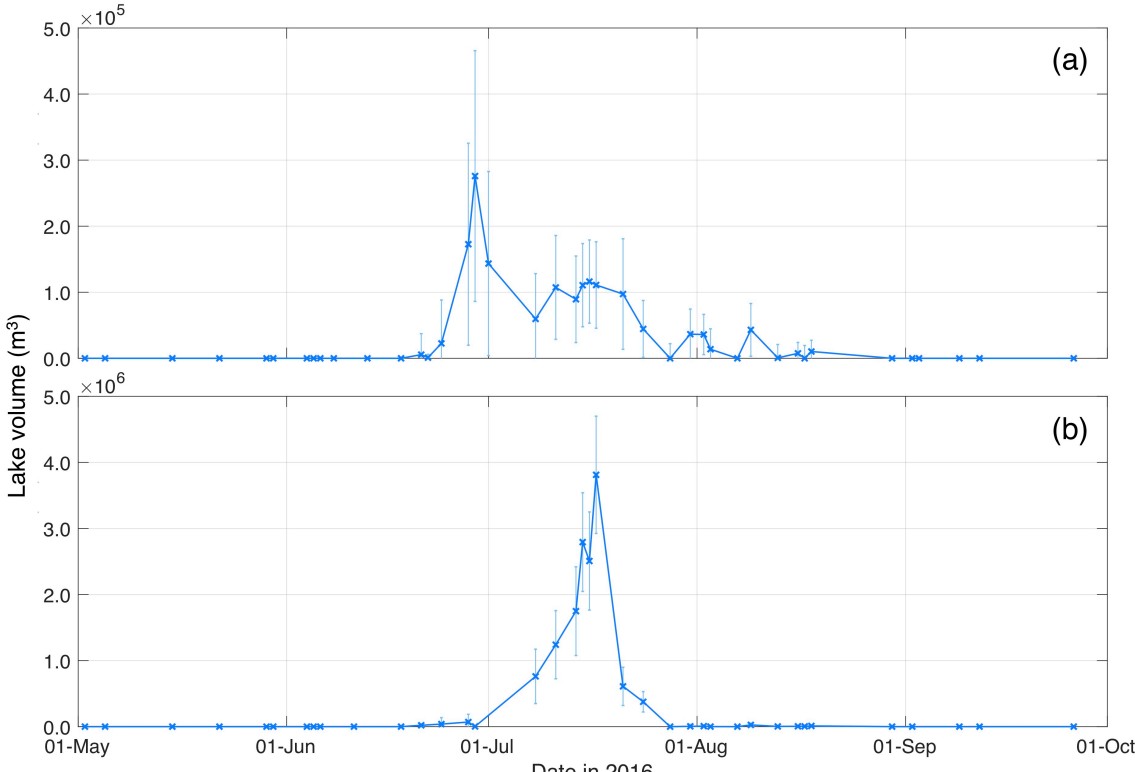

**Figure 5:** Sample time series of lake volume to show seasonal changes for (a) a non-rapidly draining lake (Fig. 1, red circle) and (b) a rapidly draining lake (Fig. 1, green box). Lines connect points without any data smoothing. Error bars were calculated by multiplying the lake-depth RMSE of 0.555 m (Sect. 3.1) by the pixel size and the number of pixels in the lake on each image.

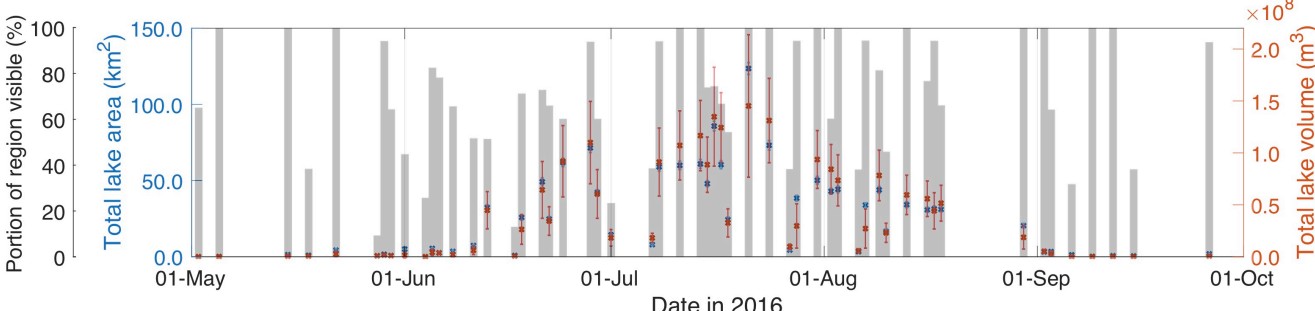

**Figure 6:** Evolution to total lake area and volume across the whole study region during the 2016 melt season. "Portion of region visible" measures the percentage of all of the pixels within the entire region that are visible in the satellite image, i.e. which are not obscured either by cloud (or cloud shadows) or are not missing data values. Figure 7 presents total lake area and volume after normalising for the proportion of region visible. Blue error bars for lake area were calculated by multiplying the lake-area RMSE of 0.007 km² (Sect. 2.3) by the number of lakes identified on each image; red error bars for total lake volume were calculated by multiplying the lake-depth RMSE of 0.555 m (Sect. 3.1) by the pixel size and the number of pixels identified as water-covered on each image.

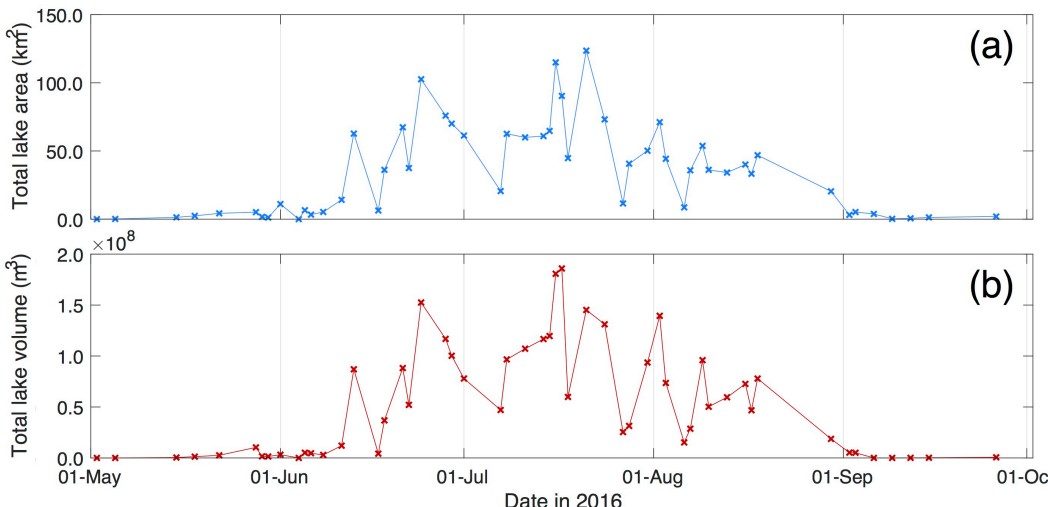

**Figure 7:** Estimates of evolution to total lake (a) area and (b) volume across the whole study region during the 2016 melt season after daily values were normalised against the proportion of the region visible of the region visible on that day (i.e. not obscured by cloud or missing data). Values are derived by dividing the daily total lake area and volume by the portion of the region visible on that day (cf. Fig. 6).

### 3.3    Rapid lake drainage

Table 2 shows the results of the identification of rapidly draining lakes using the three different datasets, and indicates that the dual-satellite record was better for identifying rapidly draining lakes than the individual records. This was for two main reasons. First, the dual-satellite record identified 118 (or 91) more rapidly draining lakes than the Landsat 8 (or Sentinel-2) record in isolation (Table 2). When either record was used alone, Sentinel-2 (or Landsat 8) performed better (or worse), identifying 50.5% (or 35.9%) of the total number of rapidly draining lakes identified by the dual-satellite record. Second, with the dual-satellite dataset, drainage dates were identified with higher precision (i.e. half of the number of days between the date of drainage initiation and cessation; Sect. 2.5.2) than with the Sentinel-2 analysis (Table 2). However, the precision appears higher for the Landsat 8 analysis than either the dual-satellite or Sentinel-2 analysis, and this is because nearly all Landsat 8 lake-drainage events occurred on two occasions when the pair of images was only separated by a day, on 8–9 July (small lakes) and 13–14 July (large lakes) (Table 2).

The dual-satellite record also identified the rapid drainage of many small lakes ($< 0.125$ km$^2$) that would not be visible with MODIS imagery due to the lower limit of its reporting size (Table 2), thus presenting an advantage of the dual-satellite record over the MODIS record of GrIS surface hydrology. These smaller lakes tended to drain rapidly earlier in the season (mean date = 8 July for the dual-satellite record) than the larger lakes (mean date = 11 July for the dual-satellite record), although the difference in dates is small, with most lakes draining in early to mid-July (Table 2; Fig. 8). In general, lakes closer to the ice margin tended to drain earlier than those inland (Fig. 8).

Finally, we tested how adjusting the thresholds used to define rapidly draining lakes would impact rapid lake-drainage identification. Changing the critical volume loss required for a lake to be identified as having drained from 80% to 70% and 90% resulted in the identification of only six more and four fewer rapid lake-drainage events, respectively. Similarly, changing the critical-refilling threshold from 20% to 10% and 30% resulted in identifying only eight fewer and five more rapidly draining lakes. However, adjusting the timing over which this loss was required had a larger impact, with adjustments from 4 to 3 and 5 days producing 37 fewer and 65 more rapid lake-drainage events, respectively.

**Table 2:** Properties of rapid lake-drainage events identified using the satellite datasets individually and when as part of a dual-satellite dataset. Large lakes are defined as $\geq 0.125$ km$^2$ (identifiable by MODIS), while small lakes are defined as $< 0.125$ km$^2$ (omitted by MODIS). "DoY" refers to day of year in 2016.

| Analysis type | Property | Large lakes | Small lakes | Total/overall |
|---|---|---|---|---|
| Sentinel-2 | Number of drainage events | 45 | 48 | 93 |
| | Percentage of total lakes | 7.5 | 8.0 | 15.5 |
| | Mean drainage date (DoY) ± mean precision | 193.4 ± 1.8 | 188.2 ± 1.6 | 190.7 ± 1.7 |
| | Minimum drainage volume ($10^5$ m$^3$) | 0.020 | 0.006 | 0.006 |
| | Maximum drainage volume ($10^5$ m$^3$) | 90.1 | 2.1 | 90.1 |
| | Mean drainage volume ($10^5$ m$^3$) | 7.5 | 0.2 | 3.7 |
| | Median drainage volume ($10^5$ m$^3$) | 1.3 | 0.2 | 0.3 |
| | Total drainage volume ($10^5$ m$^3$) | 337.3 | 11.7 | 349.0 |
| Landsat 8 | Number of drainage events | 30 | 36 | 66 |
| | Percentage of total lakes | 6.6 | 7.9 | 14.6 |
| | Mean drainage date (DoY) ± mean precision | 196.8 ± 0.6 | 190.5 ± 0.5 | 193.4 ± 0.5 |
| | Minimum drainage volume ($10^5$ m$^3$) | 0.100 | 0.050 | 0.050 |
| | Maximum drainage volume ($10^5$ m$^3$) | 19.8 | 1.1 | 19.8 |
| | Mean drainage volume ($10^5$ m$^3$) | 4.2 | 0.4 | 2.1 |
| | Median drainage volume ($10^5$ m$^3$) | 1.6 | 0.4 | 0.6 |
| | Total drainage volume ($10^5$ m$^3$) | 126.8 | 14.1 | 140.9 |
| Dual Sentinel-2 and Landsat 8 | Number of drainage events | 79 | 105 | 184 |
| | Percentage of total lakes | 11.4 | 15.2 | 26.7 |
| | Mean drainage date (DoY) ± mean precision | 193.1 ± 1.1 | 190.1 ± 1.0 | 191.4 ± 1.1 |
| | Minimum drainage volume ($10^5$ m$^3$) | 0.006 | 0.007 | 0.006 |
| | Maximum drainage volume ($10^5$ m$^3$) | 91.0 | 1.6 | 91.0 |
| | Mean drainage volume ($10^5$ m$^3$) | 7.4 | 0.3 | 3.4 |
| | Median drainage volume ($10^5$ m$^3$) | 1.8 | 0.2 | 3.9 |
| | Total drainage volume ($10^5$ m$^3$) | 586.1 | 31.2 | 617.3 |

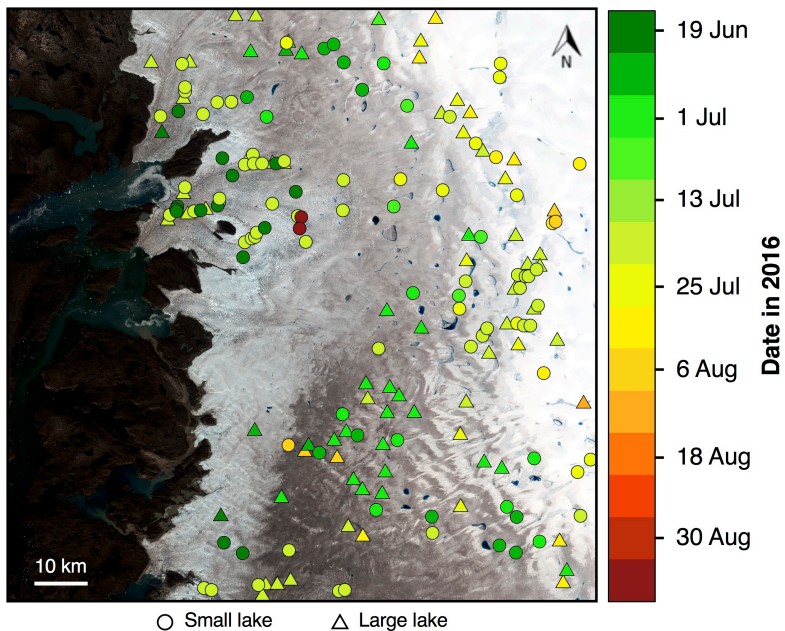

○ Small lake    △ Large lake

**Figure 8:** Dates of rapid drainage events for small (circles) and large (triangles) lakes in 2016. The panel coverage and background are the same as that shown in Fig. 1. The extreme colour bar values include those dates outside of the range shown (i.e. before 19 June and after 5 September, respectively).

### 3.4    Runoff deliveries and moulin opening by rapid lake drainage

Each rapid lake-drainage event from the dual-satellite record delivered a mean water volume of $3.4 \times 10^5$ m$^3$ (range = 0.006–$91.0 \times 10^5$ m$^3$; $\sigma = 10.2 \times 10^5$ m$^3$) into the ice sheet (Table 2; Fig. 9). Figure 9 shows the patterns of runoff delivery across the region, suggesting that small ($< 0.125$ km$^2$) and large ($\geq 0.125$ km$^2$) rapidly draining lakes were randomly distributed across the region, although there were more numerous smaller lakes at lower elevations in the north. Figure 10 shows that large lakes generally contained more water than small lakes, as might be expected, but also shows that large lakes contained a higher range of water volumes than small lakes, producing an overlap between the lake types for the lower water-volume values. Thus, although large lakes each covered a higher area, some large lakes must have been relatively shallow, as also suggested by the red to yellow coloured triangles on Fig. 9.

Using the data from the dual-satellite record, and considering just the water volumes delivered into the GrIS during lake-drainage events (and not subsequently via the moulins opened), the drainage of small ($< 0.125$ km$^2$) lakes delivered a total runoff volume of $31.2 \times 10^5$ m$^3$ into the GrIS, which is just 5.1% of the total volume ($617.3 \times 10^5$ m$^3$) delivered into the GrIS during the drainage of all lakes across the region (Table 2). Although this volume is low, small lake-drainage events, like large lake-drainage events, are additionally important because they are associated with the opening of moulins that transport surface runoff into the GrIS, and perhaps to the bed, for the remainder of the season, assuming that the moulin remains open (e.g. Banwell *et al.*, 2016; Koziol *et al.*, 2017). Associating lake drainages with moulin opening in this way means that the dual-satellite record found an additional 105 moulins (Table 2) that would not have been identified by MODIS; this is greater than the total number of moulins associated with large lake-drainage events (79) that could have been identified by MODIS, assuming MODIS can identify all lakes $> 0.125$ km$^2$, which itself is unlikely (Leeson *et al.*, 2013; Williamson *et al.*, 2017). Figure 11 shows that the moulins opened by the rapid drainage of small lakes allowed a higher total volume of runoff to enter the GrIS than that routed via moulins opened by rapidly draining large lakes; in total, moulins opened by small (or large) lakes channelled $1.61 \times 10^{11}$ (or $1.04 \times 10^{11}$) m$^3$ of runoff into the GrIS's internal hydrological system. Thus, moulins opened by small (or large) lakes delivered 61.5% (or 38.5%) of the total runoff into the GrIS after opening. Moreover, moulins opened

by small lakes delivered more runoff into the GrIS than those opened by large lakes across all ice-elevation bands, although this finding is more pronounced at lower elevations than higher elevations, i.e. below and above 800 m a.s.l. respectively (Fig. 11). The runoff into the moulins opened by small lakes also tended to reach the GrIS's internal hydrological system earlier in the season than that delivered into the moulins opened by large lakes, because these small lakes tended to drain slightly earlier (Fig. 11).

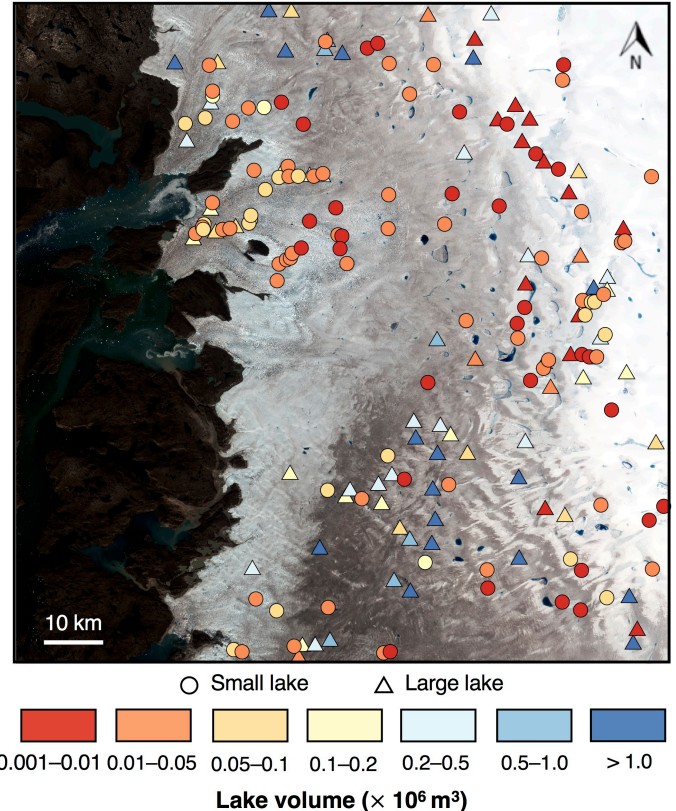

**Figure 9:** Lake water volumes measured using the physically based technique on the days prior to rapid drainage, categorised into small (< 0.125 km² in area; circles) and large lakes (≥ 0.125 km² in area; triangles). Each point shown is also assumed to represent the location at which a moulin is opened by hydrofracture during rapid lake drainage, which then remains open for the remainder of the melt season. The panel coverage and background are the same as that in Fig. 1.

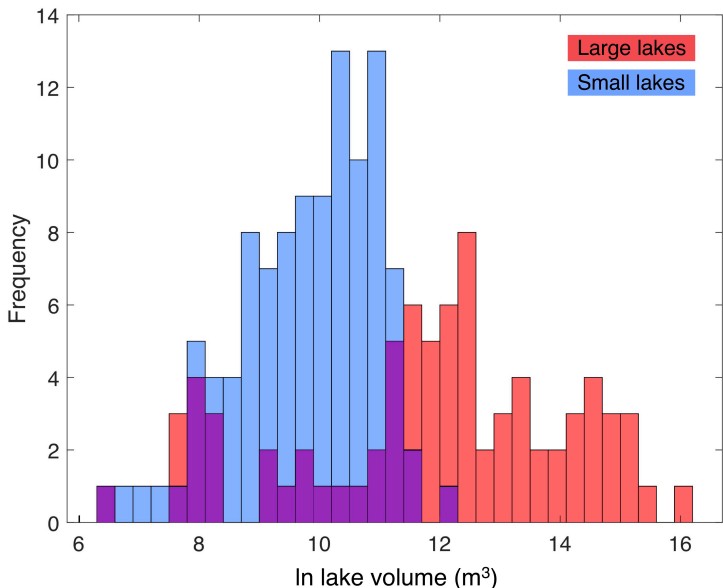

**Figure 10:** Frequency distribution of water volumes prior to rapid drainage for small and large lakes to show the lower and more tightly clustered water volumes contained within small lakes compared with large lakes. Natural logs of water volumes were taken for presentation purposes.

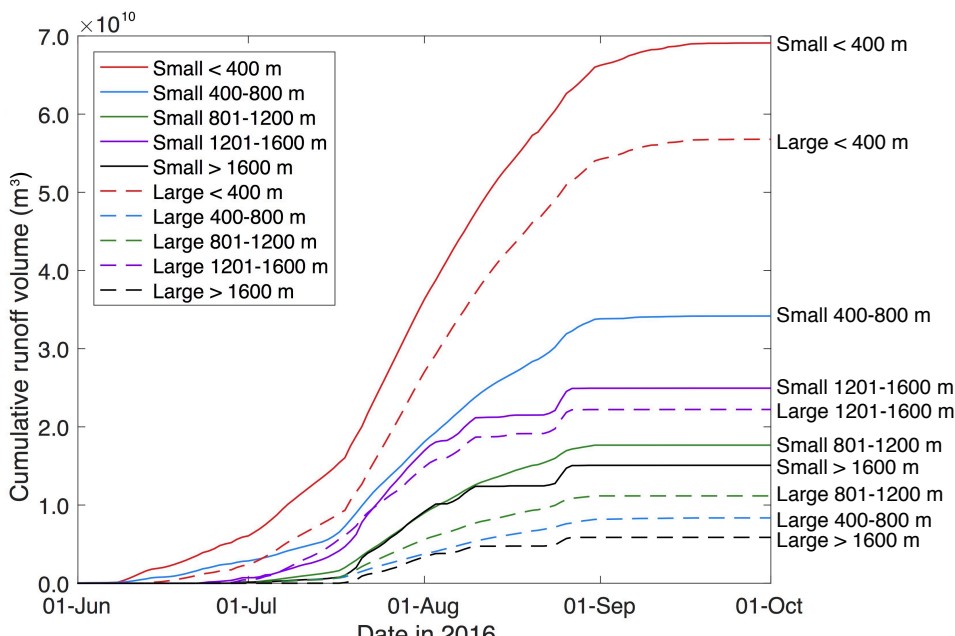

**Figure 11:** Cumulative runoff volume, from RACMO2.3p2 data (Noël *et al.*, 2018), entering the GrIS over the remainder of the melt season via the moulins opened by rapid lake drainage for small ($< 0.125$ km$^2$) and large ($\geq 0.125$ km$^2$) lakes for different ice-surface-elevation bands, derived from Howat *et al.* (2014), shown in m a.s.l. in the legend and line labels. Runoff volume is derived within the ice-surface catchments of lakes and is assumed to reach the moulin instantaneously on each day, without any flow delay.

## 4    Discussion

### 4.1    Sentinel-2 lake-depth estimates

The first and second aims of this study involved trialling and then applying a new method for calculating lake depths from Sentinel-2 imagery. We found an RMSE of 0.555 m for lake depths calculated with the physically based method applied to

Sentinel-2's red band when compared with lake depths calculated for Landsat 8 using existing methods (Pope *et al.*, 2016). When we applied the physically based method to Sentinel-2's green band and compared the depths with Landsat 8 measurements, we found a slightly lower RMSE, but the Sentinel-2 depths were unrealistically high compared with Landsat 8 values, and so this method was excluded (Table 1; Fig. S2). We selected the physical method over the empirical one because the empirical method cannot be applied without the site- or time-specific adjustments suggested in previous research (e.g. Sneed and Hamilton, 2007; Pope *et al.*, 2016; Williamson *et al.*, 2017), and might therefore perform more poorly in other years and/or for other regions of the GrIS. Given that the performance of the two methods was very similar, it therefore seemed most sensible to use the more robust physically based technique. In addition, the RMSE value (0.555 m) obtained here using the physically based method applied to the red band is only slightly higher than the error on lake-depth calculations using the physical method for similar-resolution Landsat 8 data (0.28 m for the red band and 0.63 m for the panchromatic band; Pope *et al.*, 2016). However, the RMSE on Sentinel-2 lake depths is less than half both that produced using the physically based method applied to coarser-resolution (250 m) MODIS red band data (1.27 m; Williamson *et al.*, 2017) and that produced using an empirical depth-reflectance relationship for MODIS (1.47 m; Fitzpatrick *et al.*, 2014). Therefore, using the Sentinel-2– Landsat 8 record over the MODIS one produces a much more reliable measure of lake-water depths on the GrIS because of the improved spatial resolution. The dual-satellite record is even further strengthened by its high temporal resolution, which approaches that of MODIS (Sect. 4.2).

Despite the low overall error for the physically based lake-depth calculations from Sentinel-2, we observed different performances on the two validation dates (1 July and 31 July; Sect. 3.1): the depths calculated for Sentinel-2 and Landsat 8 showed closer agreement on 1 July than on 31 July (Fig. 3). This is likely because clouds obscured a large portion of the image from 1 July. Although the lakes used for comparison were cloud-free and pixels within 200 m of a cloud-marked area were filtered, there were likely adjacency effects (at distances > 200 m from the clouds) associated with the cloud, which had more of an impact for Sentinel-2 than for Landsat 8. For example, the pixel brightness might have been reduced in locations relatively close to the clouds, consequently producing higher lake depths with the seemingly darker water (Fig. 3). Similar cloud adjacency effects have been recorded with other satellites, such as MODIS (Feng and Hu, 2016), and Landsat 8 and RapidEye (Houborg and McCabe, 2017). The depths calculated with the physical method on 31 July (Fig. 3), when there was less cloud cover, are therefore more likely to be *true* depths than the depths from 1 July when the image was affected by the cloud, even though the 1 July depths *appear* to be more correct. Assuming this is the case, our results indicate that Sentinel-2 may not accurately record deeper water (> ~3.5 m; Fig. 3) using the physical method applied to the red band. This is perhaps because the red wavelengths become saturated (i.e. fully attenuated) within the water column at higher depths, a result similar to that observed for lake-depth measurements from WorldView-2 and Landsat 8 (Moussavi *et al.*, 2016; Pope *et al.*, 2016; Williamson *et al.*, 2017). This is also likely to explain the lower maximum lake volume recorded in this study ($1.2 \times 10^7$ m$^3$) compared with previous work, including the maximum lake volume of $5.3 \times 10^7$ m$^3$ identified by Box and Ski (2007).

Alternatively, the presence of clouds on the 1 July image might be indicative of a difference in the atmospheric composition on that day, which could have affected the lake-depth calculations with Sentinel-2, but not to the same degree with Landsat 8. This might be because of the difference in bandwidths between the satellites, or because Landsat 8's panchromatic band (used for calculating lake depths) is less sensitive to the presence of clouds on an image, therefore producing more reliable lake-depth measurements. The effect of clouds on the atmosphere could have been better accounted for if the Sentinel-2 TOA reflectance data had been first converted to bottom-of-atmosphere (i.e. surface-reflectance) measurements. However, while surface-reflectance data are available for Landsat 8's optical bands, they are not for its panchromatic band, meaning that the lake-depth calculation method used here could not have been applied to generate reliable ground-truth data. We therefore intentionally chose not to perform this correction on the Sentinel-2 TOA data because we wished to directly compare the measurements from the two satellites.

Finally, in this study, the Sentinel-2 lake depths were validated using Landsat 8 measurements, which were regarded as ground-truth data, in line with a previous study involving validation of depths calculated with MODIS (Williamson *et al.*, 2017). This approach was justified since previous work (Pope *et al.*, 2016) indicated a close agreement between Landsat 8 lake depths and DEM measurements. However, it is important to note that the Landsat 8 data have errors associated with them, including a possible under-measurement of the deepest water due to saturation of the red band within the water column (Moussavi *et al.*, 2016; Pope *et al.*, 2016). Future work involving Sentinel-2 lake-depth calculations could therefore alternatively validate Sentinel-2 lake-depth estimates using different ground-truth validation data, such as higher-resolution (e.g. WorldView-2) satellite imagery, high-resolution DEM measurements of lake basins, or field lake-depth measurements.

## 4.2    Lake evolution

The second aim of this research was to apply the new methods for calculating lake areas, depths and volumes from Sentinel-2 imagery alongside those for Landsat 8 within the FASTER algorithm to produce time series for the evolution of all lakes. Applying this algorithm to the dual-satellite record allowed us to track the evolution of 690 lakes. The mean lake size (0.137 km$^2$) was just above the threshold (0.125 km$^2$) of lake size that MODIS can identify. Using a dual-satellite record, we were therefore able to achieve both high temporal (2.8 days) and spatial resolution (10–30 m). Previous studies (e.g. Selmes *et al.*, 2011, 2013; Fitzpatrick *et al.*, 2014; Miles *et al.*, 2017; Williamson *et al.*, 2017) have acknowledged that MODIS is useful because it can provide very high temporal resolution (up to sub-daily repeat site imaging) since the GrIS's surface hydrology can change quickly. However, because of the coarse spatial resolution, lake area and depth can only be calculated with large errors: for example, Williamson *et al.* (2017) calculate errors on MODIS lake areas of 0.323 km$^2$ (nearly fifty times larger than the value derived in the present study) and on MODIS lake depths of 1.27 m (twice that obtained in this study). The minor loss of temporal resolution (i.e. a reduction from daily to 2.8 days) by using the dual-satellite record rather than MODIS is therefore overcome by the record's improved reliability for resolving lake areas and depths. The use of Sentinel-2B data (available from 2017) alongside the Sentinel-2A data used here would allow further improvements to the dual record's temporal resolution; for example, in 2017, the temporal resolution of the Sentinel-2 data could be improved to an average of 1.4 days (if including all cloud-covered images) or 1.9 days (if excluding near-100% cloud-covered images) (Williamson, 2018a).

## 4.3    Rapid lake drainage

The third and fourth aims of the work were to identify the lakes tracked by the FASTER algorithm that drained rapidly, and to investigate the quantity of runoff reaching the GrIS's internal hydrological system both during the drainage events themselves and subsequently via the moulins opened by rapid lake drainage, since recent work (Banwell *et al.*, 2016; Koziol *et al.*, 2017) has shown that the moulins opened by rapid lake-drainage events allow much greater runoff volumes to reach the subglacial system than the volumes released during the actual drainage events themselves. Most research to date has used MODIS imagery to identify rapidly draining lakes because the high temporal resolution is required to separate rapidly draining lakes from those draining slowly. Although this MODIS-based research has been helpful for quantifying the characteristics of relatively large lakes ($\geq 0.125$ km$^2$) and the potential controls on their rapid drainage (Box and Ski, 2007; Morriss *et al.*, 2013; Fitzpatrick *et al.*, 2014; Williamson *et al.*, 2017, 2018), such work has been unable to study smaller ($< 0.125$ km$^2$) lakes.

Rapid drainage of both large and small lakes can be identified using the FASTER algorithm with the dual-satellite record. Although the water volumes associated with the drainage of small lakes into the GrIS amount to just 5.1% of the total water volume associated with the drainage of all lakes across the region, rapid drainage of small lakes is important because, like large lakes, they open moulins that can direct surface runoff into the GrIS's internal hydrological system over the remainder of the season. This assumes that the moulins remain open for the rest of the melt season, and we note that this may vary across the study region according to ice thickness or stress state. However, acknowledging this assumption, with the dual-satellite

record, we identified 105 small rapid lake-drainage events, thus providing 105 more input locations for surface runoff to reach the ice sheet's internal hydrological system than would be identified by MODIS. The moulins opened by small lake-drainage events are particularly important because in total they deliver over half (61.5%) of the total runoff delivered via all moulins into the GrIS's internal hydrological system. This is because the small rapidly draining lakes are more numerous (105 compared with 79) and tend to be at lower elevations than the larger lakes (small lake mean elevation = 697 m a.s.l. and σ = 514 m a.s.l.; large lake mean elevation = 848 m a.s.l. and σ = 563 m a.s.l.), where surface melting is higher. Moreover, small lakes tend to drain slightly earlier in the melt season (Table 2), meaning that the moulins they open can receive runoff for a slightly greater proportion of the melt season. In addition, the moulins opened by small lake-drainage events tend to result in higher volumes of runoff reaching the GrIS's internal hydrological system earlier rather than later in the season (Fig. 11), which may be important because the subglacial system is likely to be less hydraulically efficient at this time (e.g. Bartholomew *et al.*, 2010; Sole *et al.*, 2011; Sundal *et al.*, 2011; Banwell *et al.*, 2013, 2016; Chandler *et al.*, 2013; Andrews *et al.*, 2014). Therefore, by including these rapidly draining small lakes, the FASTER algorithm with the dual-satellite record could be used to provide a better dataset than previously for the testing of lake-filling and -draining models (e.g. Banwell *et al.*, 2012; Arnold *et al.*, 2014), or alternatively to specify the input locations and water volumes for the forcing of subglacial-hydrology models with much greater confidence than would be possible with MODIS alone. Further work is still required, however, to determine whether the water volumes delivered by these small lakes during the drainage process are capable of temporarily pressurising the subglacial drainage system, such that ice velocity speed-up events may occur, and to determine whether the associated deviations from background stresses in the far field would be sufficient to open moulins outside the basins of small lakes or to trigger chain-reaction-style rapid lake drainage (cf. Christoffersen *et al.*, 2018; Hoffman *et al.*, 2018).

Over the 2016 melt season, 27% of all lakes detected in the region drained rapidly, compared with 21% that drained rapidly in 2014 across the slightly smaller Paakitsoq region contained within the region of this study (Williamson *et al.*, 2017). However, that earlier study used MODIS imagery, so it omitted the rapid drainage of small lakes, which could explain the lower percentage if it is assumed that these small lakes are more likely to drain rapidly than the large ones, relative to the total numbers of lakes in each category. Therefore, considering just the rapid drainage of large lakes (i.e. which could be identified by MODIS) we found that 18% of large lakes drained rapidly, which is similar to the 21% value in Williamson *et al.* (2017). The 27% value in this study compares well with that of 22% from Miles *et al.* (2017), who also tracked changes to small lakes using a similar tracking threshold to that used here, albeit for a different combination of satellite platforms (Landsat 8 and Sentinel-1), and for a larger region of West Greenland in the 2015 melt season. The precision of rapid lake-drainage dates in this study (± 1.1 days) is higher than that identified by Miles *et al.* (2017) (± 4.0 days); this likely results from the different temporal resolution of Sentinel-1 compared with Sentinel-2, and because Miles *et al.* (2017) were forced to discard some images before conducting their analysis, reducing the average temporal resolution and so the ability to identify rapid lake drainage dates confidently. Thus, although Sentinel-1 can image through clouds, the Sentinel-1 record suffers from separate issues that offset this advantage. The shorter time interval for repeat lake imaging offered by the FASTER algorithm is likely to help reduce the observation bias associated with the longer time intervals of existing remote-sensing (e.g. MODIS-based) work (Cooley and Christoffersen, 2017). Finally, we also offer an advance with the dual-satellite method since we can calculate the water volumes delivered into the GrIS by rapid lake drainage (because of the use of optical satellite data), in addition to the lake-area changes that could be tracked previously (e.g. Miles *et al.*, 2017).

## 5    Conclusions

We have presented the results of the first approach to combine two medium-resolution optical satellite datasets (Sentinel-2 and Landsat 8) to generate the highest spatial- and temporal-resolution record of lake area and volume evolution on the GrIS to date. To achieve this, we have exploited the increasing availability of medium- to high-resolution satellite imagery, and then

combined these newly available data with recent techniques for automatically tracking changes to lake areas and volumes, and for identifying rapid lake drainage. The resultant FASTER algorithm allows lake areas and volumes to be calculated with high accuracy from Sentinel-2. For lake area, the RMSE is 0.007 km$^2$ when compared with that derived from Landsat 8 data, which is nearly fifty times lower than the error associated with MODIS. For lake depth, the RMSE is 0.555 m, under half that associated with MODIS. The techniques for lake-area and lake-depth calculation from Sentinel-2, when combined with similar techniques applied to Landsat 8 data, yielded a dual-satellite record with comparable temporal resolution (2.8 days) to that of MODIS (daily). Thus, the FASTER algorithm applied here reduces the large errors associated with calculating lake depth and lake area using the ancestral FAST algorithm applied to the coarser spatial resolution MODIS imagery (Williamson *et al.*, 2017). In addition, the FASTER algorithm provides a similarly frequent site revisit time as MODIS, allowing rapid lake drainage to be identified with high precision (± 1.1 days). Our work shows that using both sets of high-resolution satellite imagery together provides better insights into lake filling and drainage than using either one in isolation. With the availability of Sentinel-2B data from summer 2017 to supplement the Sentinel-2A data used in this study, the three datasets could be used together to generate an even higher temporal resolution record. In the future, the dual-satellite record presented here is therefore likely to be able to replace, or at least supplement, the MODIS record used to investigate lakes on the GrIS.

We have additionally taken advantage of new, and increasingly reliable, downscaled regional climate-model (RACMO2.3p2) output data (Noël *et al.*, 2018) to provide insights into the runoff volumes entering the GrIS's englacial or subglacial hydrological systems after moulin opening was identified using the FASTER algorithm. Our results show that the water volumes released into the GrIS by small lakes during the lake-drainage events themselves are small (only 5.1%) relative to the volumes released by all lake-drainage events, suggesting small lakes are less important in this sense. However, of the total water volume that subsequently reaches the GrIS's internal hydrological system via all moulins opened by lake drainage (from both large and small lakes), moulins opened by small lakes deliver 61.5% of the total water volume. This suggests that small lakes are important to include in future remote-sensing and modelling studies.

Resulting from the above, the FASTER algorithm holds great potential for generating novel insights into lake behaviour on the GrIS from remote sensing, including for small lakes that change quickly (cf. Miles *et al.*, 2017). Future work should focus on applying the FASTER algorithm to wider areas of the GrIS and comparing the results with increasingly available and reliable high temporal resolution ice-velocity data (e.g. Joughin *et al.*, 2018) to investigate the influence of lake drainage on the observed patterns of intra- and inter-annual velocity variations across the GrIS. Moreover, the high spatial resolution record could be used to identify the potential controls on the initiation of rapid lake drainage, something that could not be achieved with MODIS data, perhaps due to the data's coarse spatial resolution (Williamson *et al.*, 2018). Finally, the water volumes delivered into the GrIS during the rapid lake-drainage events identified with this record, the moulins that are assumed to open during such events, and the subsequent runoff that enters the GrIS via these moulins, could be used as forcing or testing data for subglacial-hydrology models (e.g. Hewitt, 2013; Banwell *et al.*, 2016) and linked hydrology-ice dynamics models (e.g. Koziol and Arnold, 2018). Ultimately, applications of the FASTER algorithm such as these could enable the GrIS's supraglacial and subglacial hydrology to be modelled more accurately in order to provide better constraints on future runoff, ice discharge and sea-level rise from the GrIS.

*Data availability.* All satellite imagery and regional climate-model output data used in the analysis are open access. The full MATLAB source code for the FASTER algorithm used to process and analyse the imagery is freely available for download (Williamson, 2018b).

*Author contributions.* AGW conceived the study, designed and executed the method presented in the research, conducted the analysis, and drafted the original manuscript, all under the supervision of the other authors. All authors discussed the results and contributed towards editing the manuscript. AGW revised the manuscript following reviewer and editorial comments.

*Competing interests.* The authors declare that they have no conflict of interest.

*Acknowledgements.* AGW was funded by a UK Natural Environment Research Council PhD studentship (NE/L002507/1) awarded through the Cambridge Earth System Science Doctoral Training Partnership and a Cambridge Philosophical Society Research Studentship. AFB was funded by a Leverhulme/Newton Trust Early Career Fellowship (ECF-2014-412). The Scott Polar Research Institute's B. B. Roberts Fund and the Cambridge Philosophical Society provided funding for AGW to present

this research at the European Geosciences Union General Assembly 2018. We are grateful to Allen Pope for discussing the results of the Sentinel-2 lake-depth calculations with us, and to Brice Noël for speedily providing the RACMO2.3p2 data. Katie Miles and Corinne Benedek are thanked for generally contributing to the idea for the study, and we thank Gareth Rees and Pete Nienow for providing thoughtful feedback on this work. Finally, detailed reviewer comments from Allen Pope, Samuel Doyle and Kristin Poinar, and editorial comments from Bert Wouters, helped to significantly improve the quality of

the manuscript.

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
