# Peer review of "Dual-satellite (Sentinel-2 and Landsat 8) remote sensing of supraglacial lakes in Greenland"

_The Cryosphere, 2018_

## Referee Comment (RC1) · A. Pope (Referee) · 20 May 2018

"Dual-satellite (Sentinel-2 and Landsat 8) remote sensing of supraglacial lakes in Greenland" by Williamson et al. explores a new method of retrieving supraglacial lake depth from Sentinel-2 imagery, combines it with Landsat 8 to build a higher-temporal resolution record, tracks lake volume/filling/draining, and investigates the impact of lakes of various size on the hydrology of the Greenland Ice sheet.

Williamson et al. have produced a paper which is clear, clean, logical, well-written, and ultimately enjoyable to read. Thoughtful consideration has been given to how to combine datasets and how to interpret the resultant data. However, there are some crucial factors that I believe the paper should consider before being published.

[Figure]

Choosing the Sentinel-2 Method:

This comparison is a big step in the paper and will facilitate many future studies. However, I think there are one or two options which really need to be carefully considered before claiming victory – in particular exploring the use of the S2 Green Band. In Figure 3, evidence of saturation is clearly evident and you note in the discussion around line 465 that this could be related to the use of the red band. So I don't understand why you do not explore using the Green band on its own, or like the L8 method, in cooperation with the Red Band? In addition, Figures 3 and 4 (and other similar) would benefit from using heat maps rather than small dots; the data density is too high for interpretation in this format. Using a 1:1 line (or similar) might also help in interpretation.

Analysis: Error & How Many Lakes?

This study would be more robust if a little more attention was added to areas that help contextualize the data. In particular: *Adding any error bars on values which are calculated for area / volume *For volume (e.g. Line 334), a 10% disagreement between S2 and L8 seems pretty good. However, a big factor that has the potential to be quite variable between image resolutions, is the calculation of the lake shore and therefore lake bottom albedo. Did you explore this effect at all? Or at least it might be good to include it in the discussion? *In Line 330, you observe more lakes in the dual record than in L8 or S2 individually. Why isn't there more match-up of the lakes being tracked in the individual datasets? Is this a result of the higher temporal density? Cloud cover differences? Other? I'm curious because the tests in the Supplement seem to so such close agreement in lake areas being measured.

The Role of Opened Conduits:

There is an assumption that one lakes drain quickly (by opening a conduit) that these conduits continue to remain open. However, there is no discussion about whether is assumption is theoretically sound or observationally verified. Is this equally likely for small and large lakes at various ice sheet thicknesses / stress states? There seem

to be a lot of variables, and this may indeed be valid, but it should be explained – in particular given the conclusions related to the role of small lakes connecting the supraglacial and subglacial hydrological systems.

Other minor comments:

Line 8: Landsat is still what I would call medium resolution, especially with a new era of sub-metre sensors. Maybe fix by changing "high spatial resolution" to "higher spatial resolution?

Line 131: Have you considered providing tables in txt form, too (or perhaps provided with code to batch download) to facilitate reproducibility?

Line 141: You write that tiles were reprojected. How were data interpolated– NN, bilinear, other? These details were carefully described in other steps, so I ask mostly for completeness.

Line 148: For those less familiar, perhaps write "band-6" as "SWIR/band-6" or similar? Here and elsewhere.

Line 151: For cloud shadow – I don't know that this is enough to handle shadow. Did you check any math / assumptions (based on cloud elevation, solar angle, etc.) that this would really be sufficient? It could be a big deal, especially for false-positives on fake drainage. Again, I'm really unsure and haven't run the numbers myself, but it seems important enough to confirm.

Line 154: For lack of a better place to put this: I believe that L8 and S2 are orthorectified using different DEMs, so there could be (slight) offsets in lake locations. Have you considered this effect or the magnitude in it? The Kaab/Paul 2016 papers you referencing think about implications for velocity tracking, but I'm just not sure of the impact in this part of the world.

Line 159: You use the cloud masks provided with the data, but is there any evidence on reliability over snow and ice?

Figure 2: Reprojecting step not included? Figure 2: I believe lake area masks should feed into lake depth calculation (e.g. finding lake edges)?

Line 180: The resampling and NDWI steps appear to be described in Figure 2 in the opposite order?

Line 231: Not an issue – I'm mostly just curious – why you chose NN here (and bilinear elsewhere)?

Line 241: Well done with these acronyms ïĄŁ

Figure 3 & 4: Include legend so figure is easier to read

Line 336: Are these symmetrical distributions? Perhaps using Quartile1 / Quartile3 info will help describe the data while also being sure to use non-parametric statistics.

Line 341: Did you ever compare with MODIS to make that the *it* failed the test (as it would be expected to)?

Figure 6/7: I really like how you can display the data in the context of which data is available!! Have you considered if there was any variability in lake distribution / more nuanced than just scaling by cloud cover percent? Like an elevation-dependent extrapolation, or something like that? Also: consider combining into one figure

Figures 8 & 9: Combine into one figure

Fig 11: These colors are not necessarily all distinguishable to a color-blind reader. Revise colors and/or label the lines on the righthand axis.

Line 578: What about sharing your resultant lake dataset? Line 578: The Cryosphere data policy encourages a few things that are different from how this paper handles sharing – most importantly share data AND code in an open, citable repository. Requiring to ask an author is a large barrier particularly in the future. See Gil et al. for further suggestions, and I'm sure that your readership would love to have these tools and dataset available in an open place / GitHub repo.

https://agupubs.onlinelibrary.wiley.com/doi/full/10.1002/2015EA000136

---

## Referee Comment (RC2) · S.H. Doyle (Referee) · 12 Jun 2018

Williamson et al. present a new technique which combines two different types of satellite imagery to create a lake geometry time series with an unprecedented combination of spatial and temporal resolution. The methods are rigorous and described in comprehensive detail. Indeed the detailed description of the methods gives the impression that the manuscript is largely a techniques paper despite significant results also being presented. The figures and tables are of a high standard and the referencing is appropriate throughout. The manuscript reads well with very few errors. The finding that a large proportion of surface water drains through small (and often previously undetected) lakes is significant. The paper builds on previous work on supraglacial lakes on the Greenland ice sheet.

**General comments**

My main concern stems from L213: *"We treated these Landsat depths and volumes as ground-truth data as in Williamson et al. (2018)"*. This assumption is not discussed in detail at any point in the manuscript, although I guess it may have been described in the author's previous paper. In my opinion, there should be more discussion in this manuscript of the lack of actual ground truthing and how this might affect the absolute accuracy of the results. That is, how do the authors expect the results to compare with reality? How does this study compare to previous studies which employed ground truthing? To be clear, I'm not expecting ground truthing to be undertaken, but the lack of ground truthing and its potential effects should be discussed.

There is also a lack of discussion of the depth limitation of the techniques used. The techniques presented only retrieve lake depth up until a certain point. Figures 3 and 4 suggest a 6 m limit, when lakes are known to be often deeper than 10 m. If many lakes are deeper than the limit then this would hypothetically create a bias in the results by underestimating the volumes of the deeper lakes, which could affect the reliability of the main conclusion of the paper — that the water drained through small lakes is greater than that through large lakes. Given this, the relatively low maximum lake volumes (1.2 x $10^7$ m$^3$) reported on L334/335 compared to those in the literature (e.g. Box and Ski, 2007) are potentially concerning. These limitations should be discussed to make the reader aware of them. The authors could also look to previous studies, which have measured lake depths to understand whether the maximum depth of their technique is potentially too shallow.

These issues aside, the discussion of problems and limitations in Section 4.1 is well-considered and appropriate.

**Specific comments**

L24/25 — consider offering an explanation for why small lakes drain such a large proportion of the runoff. Is it because they are more numerous? An explanation for this

observation is never given in the manuscript.

L29 — the last two references in this sentence are in the wrong place. They currently only relate to there being two main ways the ice sheet loses mass (without specifying what they are), which as a statement does not need a citation. Perhaps move them to later in the sentence/paragraph.

L40 — I'm not sure whether the citations to Doyle et al. (2018) and Hofstede et al. (2018) are appropriate here. Presumably they are given here as evidence for subglacial sediment? In fact the number of citations given here could be significantly reduced to list only the key studies.

L43 — Consider the recent paper by Christoffersen et al. (2018) here.

L44 — It is well established that surface water delivered to the bed of the Greenland ice sheet accelerates ice flow in the short term. Given the evidence for this, the phrase 'potentially explaining' seems a bit weak.

L45 — Consider citing Joughin et al. (2013) and Hoffmann et al. (2011) here.

L75 — I'm not aware of any evidence which suggests lake hydrofracture takes days. All the evidence suggests the process is rapid, taking hours or less.

L111 — May to October is not summer, perhaps use 'melt season' instead, although there is not usually much melt in October.

L124 — Why does the study only cover up to 90 km inland? Perhaps give a reason here.

L187 — stating the increments provides no information on how much it was adjusted in absolute terms, unless only one increment was used? Or is the percentage change given per 0.01 increment? Please clarify.

L230 — Can you (briefly) be a bit more specific here as to how the new technique was validated, even if it requires some repetition.

L260 — Consider discussing the implications of Cooley and Christoffersen (2017) regarding the effects of observation bias on the detection of rapidly draining lakes. The reduced interval of the new technique presented in the manuscript under review should reduce the observation bias associated with the longer intervals of previous studies.

L271 and L418 and L496 and elsewhere — The term 'interior' is here used as a synonym for 'englacial'. This is arguably ambiguous with the frequent use of ''interior' to describe the central region of the ice sheet (including the surface) away from the margin. The use of this term here also neglects the important effects of surface water reaching the bed.

L335 - These maximum volumes seem quite low. How do they compare with the literature? Are there any reasons why the volumes are low? Is it due to the depth limitation of the techniques used?

L411 — The two modelling studies cited here were not the first to suggest this. Consider citing other studies and/or adding an e.g. before the citation.

L418 — Why is this? Why does more water drain through small lakes than larger ones? Can it be explained by the greater number of small lakes, or is it a result of drainage basin size, or is it a result of a bias in the technique? Some discussion is warranted.

L503 and other occurrences of this pair of citations — these modelling studies may not be considered as the first or most appropriate for the establishment of drainage through moulins, which has been known for a long time (and was not determined by modelling). Consider listing earlier citations and/or giving an e.g. first to show that these citations are selected examples. Some of the sentences preceding these citations (including that on L503) may not even need a citation.

L511 — 'runoff' not 'meltwater'.

L512 — '*the* moulins'.

Fig. 1 — Consider labelling some glaciers to aid the reader. Also, consider showing

enlarged images of each of the two example lakes as subplots to demonstrate the capability and resolution of the imagery.

Fig. 3 — at this scale, whether the markers are circles or squares is redundant.

Fig. 10 — consider a red/blue transparency with purple overlap (or any other primary colour pair).

**Technical corrections**

L20 — 'identify' is the wrong word here, suggest 'estimate'

L25 — strictly speaking its not via *all* moulins but only those identified within 'small' and 'large' lakes.

L95 — define MSI.

L112 — the last sentence of this aim isn't written as an aim and there is change of tense from the adjacent sentences.

L114 — define 'rapidly' here.

L208 and other occurrences — within the text write out 'Section' or 'Figure' in full.

L258 — citation should be to Doyle et al. (2013).

L297 — replace 'more poorly' with 'worse'.

L330 — what does the number in brackets refer to?

L376 — Suggest: 'when the pair of images were only separated by a day'.

L391 — "Large lakes are *defined as* . . . "

L396 and L405 and L428 — Write out 'Figure' within text.

L434 — consider rearranging to avoid double brackets.

L465 —'entirely' is not necessary here.
L490 — 'offset' is the wrong word here.

Table 2 — Consistency with precision. Specifically, minimum drainage volume should be given as 0.020 for large lakes for Sentinel 2 (not 0.02). The same applies to the same for Landsat 8.

Section S1 — delete 'the value of' in the first sentence.

Table S2 — Write out 'The asterisk denotes . . .'

Fig. S1 — Overlap in X-axis label superscript. Also, consider inserting 'therefore' in '. . . between the two sets of lake areas is *therefore* remarkably small'.

**Additional references (not already cited in the manuscript)**

Christoffersen, P., Bougamont, M, Hubbard, A., Doyle, S.H., Grigsby, S. & Pettersson, P. 2018. Cascading lake drainage on the Greenland ice sheet triggered by tensile shock and fracture, *Nature Comms.*, 9 1064.

Cooley, S. W., & Christoffersen, P. (2017). Observation bias correction reveals more rapidly draining lakes on the Greenland Ice Sheet. *Journal of Geophysical Research: Earth Surface*, 122, 1867—1881

Joughin, I., Das, S., Flowers, G., Behn, M., Alley, R., King, M., Smith, B., Bamber, J., van den Broeke, M. & van Angelen, J. 2013. Influence of ice-sheet geometry and supraglacial lakes on seasonal ice-flow variability, *The Cryosphere*, 7, 1185-1192.

---

## Referee Comment (RC3) · K. Poinar (Referee) · 15 Jun 2018

**Summary**

Williamson et al describe a substantial new contribution to the remote-sensing detection of Greenland supraglacial lake drainage events. Their approach is to combine images from two medium- to high-resolution sensors to achieve near-daily time resolution of a study area in WNW Greenland. The authors apply this new technique over the 2016 melt season and are able to detect smaller draining lakes ($<0.125$ km$^2$) than previously possible, with good temporal precision ( $\pm \sim 1$ day). By combining the new lake drainage dataset with regional climate model output analyzed across surface catchments, the authors conclude that smaller ($<0.125$ km$^2$) fast-draining lakes, which

previous coarser analyses have missed, actually contribute a majority of lake water to the subglacial system across the study area. Their result shows the importance of using high-res techniques, such as presented here, to identify the locations and timing of lake water input to the subglacial system.

**Specific comments**

The methods portion of the paper is thorough and appears to be robust; the study uses previous work (by Pope and by Williamson) to validate its new methods.

In the results portion, it was unclear which data were used to develop the empirical relationships between L8 lake depth and S2 TOA reflectance (July 1 / 31?), and which data were used to evaluate it (Table 1; all image dates?). Perhaps the distinction is not of great importance (I cannot tell), but this could be easily clarified.

I would suggest against the use of the word "error" in lake-drainage dates. "Error" suggests that the true dates of lake drainage are known; however, they are not known. "Precision" would thus be a more accurate term.

Given the new ability to precisely identify drainage dates of more lakes than ever before, I read the Discussion with great interest. I think two of the inferences made in this section were a bit weak. However, these were not the main contribution of the work, and so scaling them back will not make a great loss to the paper.
1. Lake size and lake drainage date – The authors attribute the high water contribution (61.5%) by small lakes to the fact that they drain earlier in the melt season than large lakes (lines 511-512). While the data in Table 2 do show a significant difference (non-overlapping date ranges) between small and large lake drainage dates, this difference (just 1-2 days) is not substantial within the context of the melt season and the evolution of subglacial hydrology.
2. Lake size and elevation – The authors also state that the lower elevations of small lakes may contribute to their greater water contribution (lines 512-513). These data do not appear in Table 2 (I would suggest adding it: the mean and std elevations of large
and small lakes), and this statement contradicts another statement (lines 400-402) that lake size and elevation are uncorrelated.

What, then, can explain the high (61.5%) contribution of the small lakes? Is it simply a large total size of their basins? This information would be easy to include (I believe it is already calculated).

Finally, and most crucially, the conclusion that small lakes are important to the subglacial hydrological system is based on the assumption that their moulins stay open for the entire melt season (Figure 11). I don't have any especial reason to doubt this, but the assumption is not backed up in the paper although I believe the authors' data could easily do so. Presumably, if a moulin were to close up before the end of the melt season, a lake would re-form on site, and could be seen in the data. I think I can infer from the description of the FASTER algorithm that any such lakes would not meet the criteria for "fast-draining" and thus would not be included in this study – but this is not stated/explained in the manuscript.

**Technical comments**

Line 7 - I wouldn't start the abstract with "Although"; move it to the middle of the sentence as "however".

Lines 47, 49 - It is not sensible heat, but latent heat that makes both of these effects (Phillips and Mankoff references). Water temperature is not important.

Line 49 - It's actually Poinar et al. 2016 (not 2017)

Line 67 - Replace "this" with its antecedent (since it is the first sentence of the paragraph).

Line 82 - The records have no problems; instead, perhaps the methods have shortcomings.

Lines 85 and 95 - Define or remove SAR, MSI acronyms

Line 135 - Clarify year 2016

Lines 155, 159 - I got a bit confused with the numbers here, since they are similar (38 + 39 = 77). Perhaps recast the sentences to use only the number 39. Also please state the year 2016 again here.

Line 190 - $R^2$ = 0.999, that's excellent, good for you!

Line 271 and elsewhere - "GrIS interior" confused me; to me it means inland or upstream regions, whereas you intend to say that the water leaves the surface and enters the englacial or subglacial environment.

Line 294 - Here you say ∼3 meters but later (line 465) you say ∼3.5 meters.

Lines 347-350 - This sentence would benefit from parallel construction.

Line 353 - p=0.00 should be more precise.

Line 443 - "We opted for" sounds a bit informal.

Lines 458-468 - Effect of July 1 "cloud adjacency". Pixels 200 m from clouds were already removed (Data and methods section), so it seems to me that this cloud-adjacency argument would not apply. Perhaps more description of how "adjacent" (i.e., if there are effects >200 m away) these effects are is required.

Lines 536-539 - It reads a little harsh on Miles et al. to end the paragraph with the shortcomings of that study; instead wouldn't it be better to end by highlighting the strengths of your own study?

Lines 540-541 - As written, this sentence is false because other studies have combined two optical satellite datasets (e.g. MODIS and L8). Recast by adding "medium-resolution" or moving the sensors out of parentheses.

Line 561 - "less" instead of "not"

Figure 7. I like that Figure 6 was scaled up to the full image region here. The data

[Figure]

show a lot of variability (sawtooth-like) on multi-day scales. I'd be interested in whether this is "real" (perhaps regionally linked lake drainages?) or just noise remaining from the effects of clouded-over regions.

Figure 10. The magnitude of lake volume (x axis) seems much too large: the largest lake would have a volume of $10^{16}$ m$^3$, which would be 100 km x 100 km x 10 km, way too big. There must be an error here.

Figure 11. The y axis label is confusing: volume, yet mm?

Table 2. Consider adding the mean surface elevation of the 3 classes of lakes, as described in the "Specific comments" section of this review.

---

## Author Comment (AC1) · 1 Aug 2018

The comment was uploaded in the form of a supplement:
https://www.the-cryosphere-discuss.net/tc-2018-56/tc-2018-56-AC1-supplement.pdf

---

## Author Response (AR1)

**Author response to reviewers for manuscript TC-2018-56**

We are very grateful to all three reviewers for their detailed and constructive reviews of our manuscript.

Below, we have responded to each of the reviewers' comments sequentially. Here, the original reviewers' comments are in normal font and our responses are in *italicised* font. At the end of this response, we have also included an updated version of the manuscript with all of the tracked changes visible.

**Response to reviewer #1 (Allen Pope)**

"Dual-satellite (Sentinel-2 and Landsat 8) remote sensing of supraglacial lakes in Greenland" by Williamson et al. explores a new method of retrieving supraglacial lake depth from Sentinel-2 imagery, combines it with Landsat 8 to build a higher-temporal resolution record, tracks lake volume/filling/draining, and investigates the impact of lakes of various size on the hydrology of the Greenland Ice sheet.

Williamson et al. have produced a paper which is clear, clean, logical, well-written, and ultimately enjoyable to read. Thoughtful consideration has been given to how to combine datasets and how to interpret the resultant data. However, there are some crucial factors that I believe the paper should consider before being published.

We are grateful of the positive comments on our manuscript, thank you.

Choosing the Sentinel-2 Method:

This comparison is a big step in the paper and will facilitate many future studies. However, I think there are one or two options which really need to be carefully considered before claiming victory – in particular exploring the use of the S2 Green Band. In Figure 3, evidence of saturation is clearly evident and you note in the discussion around line 465 that this could be related to the use of the red band. So I don't understand why you do not explore using the Green band on its own, or like the L8 method, in cooperation with the Red Band?

Thanks for the suggestion. We had previously conducted the analysis for the green band but chose to exclude it from the manuscript because the depths that were produced for Sentinel-2 were much too high relative to those calculated for Landsat 8: for example, maximum predicted lake depths from Sentinel-2 were ~19 m, compared with values of only ~5.5 m in the Landsat 8 data. However, we have now decided to include this analysis in the paper as a supplementary figure (and with details of the derived relationship in text and in Table 1), in a similar way to the empirical relationships as applied to the green-band and blue-band data, as we think that it will be helpful for other workers in the future. Given that there was such over-prediction by the green band, we chose not to conduct any joint analysis involving using both the green and red bands. We do think that future work would be merited, however, to help overcome the issue of saturation in the Sentinel-2 red band technique, but we are aware that this is an issue that also exists for Landsat 8, as documented in previous work.

In addition, Figures 3 and 4 (and other similar) would benefit from using heat maps rather than small dots; the data density is too high for interpretation in this format. Using a 1:1 line (or similar) might also help in interpretation.

Thank you for the suggestion, but we think that a heat map format would not work well here, because we are trying to display two sets of data on a single figure, and as it stands, the data points are displayed in a semi-transparent way to show the locations on the plot where there are more data points. This means it is possible to see where the data overlap from different dates, while ensuring that all of the data are still visible. Thus, we think that the figure formats we have chosen work in a similar way to a heat map but also allow two sets of data to be displayed at once. However, we have added a 1:1 line to aid with interpretation, as requested, as we think that this is very helpful.

**Analysis: Error & How Many Lakes?**

This study would be more robust if a little more attention was added to areas that help contextualize the data. In particular:

• Adding any error bars on values which are calculated for area / volume

**Apologies, we overlooked including them. Added.**

• For volume (e.g. Line 334), a 10% disagreement between S2 and L8 seems pretty good. However, a big factor that has the potential to be quite variable between image resolutions, is the calculation of the lake shore and therefore lake bottom albedo. Did you explore this effect at all? Or at least it might be good to include it in the discussion?

This was something we did briefly explore in the earlier analysis (for scene-by-scene lake-depth calculations), and the result for Sentinel-2 was including a lake shore comprising the lake-area mask dilated by two pixels (i.e. 20 m) instead of a single pixel (i.e. 10 m). We have now included an extra clause here to explain that it is crucial to ensure that the lake-bottom albedo values are correctly defined given the sensitivity of the method to this parameter, and this is why we chose to include two pixels around the lake edge.

In Line 330, you observe more lakes in the dual record than in L8 or S2 individually. Why isn't there more
match-up of the lakes being tracked in the individual datasets? Is this a result of the higher temporal density?
Cloud cover differences? Other? I'm curious because the tests in the Supplement seem to so such close
agreement in lake areas being measured.

The comparisons of lake area presented in the Supplement were only for lakes that were present in both the Landsat 8 and Sentinel-2 datasets, which explains the close agreement between the two sets of values. The reason that more lakes are present in the dualsatellite record than either record individually is because of the dual-satellite record's higher temporal resolution. This results in a greater number of pixels across the whole image stack being marked as water-covered using the dual-satellite record at least once in the season, and thus being included in the calculations of maximum lake extents across the whole season. When we then filter out groups of pixels that do not reach more than 495 pixels at least once in the season (see Sect. 2.5.1), more groups of pixels meet this size with the dual-satellite record than with either record individually, and so more lakes are tracked. We hope that this makes sense.

**The Role of Opened Conduits:**

There is an assumption that one lakes drain quickly (by opening a conduit) that these conduits continue to remain open. However, there is no discussion about whether is assumption is theoretically sound or observationally verified. Is this equally likely for small and large lakes at various ice sheet thicknesses / stress states? There seem to be a lot of variables, and this may indeed be valid, but it should be explained – in particular given the conclusions related to the role of small lakes connecting the supraglacial and subglacial hydrological systems.

We agree that this is indeed an assumption with this method. We have now chosen to acknowledge that there is an assumption that the moulins will remain open more clearly at several places in the manuscript, including in the discussion, where we have stated that this assumption may vary across the study region according to ice thickness or stress state, for example.

**Other minor comments:**

Line 8: Landsat is still what I would call medium resolution, especially with a new era of sub-metre sensors. Maybe fix by changing "high spatial resolution" to "higher spatial resolution?

**Done.**

Line 131: Have you considered providing tables in txt form, too (or perhaps provided with code to batch download) to facilitate reproducibility?

Full source code for the FASTER algorithm is now available online publicly to facilitate reproducibility; please also see response to the later comment on the same topic of distributing data.

Line 141: You write that tiles were reprojected. How were data interpolated – NN, bilinear, other? These details were carefully described in other steps, so I ask mostly for completeness.

**Thanks for noting the omission. This was bilinear interpolation – now clarified in text.**

Line 148: For those less familiar, perhaps write "band-6" as "SWIR/band-6" or similar? Here and elsewhere.

**Done here and elsewhere.**

Line 151: For cloud shadow – I don't know that this is enough to handle shadow. Did you check any math / assumptions (based on cloud elevation, solar angle, etc.) that this would really be sufficient? It could be a big deal, especially for false-positives on fake drainage. Again, I'm really unsure and haven't run the numbers myself, but it seems important enough to confirm.

That was unclear, apologies. Images were manually verified for shadowing. This has been clarified in text. The 200 m value was chosen based on manually inspecting images with the thresholds applied, and occasionally some finer clouds near to the edges of larger clouds had been missed with the classification, and a 200 m radius helped to remove them. An alternative automated approach would be to include a filter based on the temporal consistency (see Williamson et al., 2017 for an example) to separate lakes (which are more persistent) from shadows (which are less persistent), but we did not see that shadowing was a large issue in this study, so we chose not to do this.

Line 154: For lack of a better place to put this: I believe that L8 and S2 are orthorectified using different DEMs, so there could be (slight) offsets in lake locations. Have you considered this effect or the magnitude in it? The Kaab/Paul 2016 papers you referencing think about implications for velocity tracking, but I'm just not sure of the impact in this part of the world.

*True – we have now mentioned this in the manuscript. We manually checked for lake offset on two contemporaneous image pairs (1 July and 31 July 2016), but did not note any obvious offset.*

Line 159: You use the cloud masks provided with the data, but is there any evidence on reliability over snow and ice?

This step was only an initial means to filter images that were likely to be particularly cloudy. Before an image was excluded entirely, it was manually checked, to ensure that the cloud mask provided with the data had produced a reliable measure of cloudiness. The actual cloud-masking steps were different (as described later in that section).

Figure 2: Reprojecting step not included?

We think that this is quite difficult to include on the figure without making it more confusing, but we have now explained in the figure caption that the steps referenced apply once the reprojection has been completed. Hopefully this is satisfactory.

Figure 2: I believe lake area masks should feed into lake depth calculation (e.g. finding lake edges)?

That's correct, thanks for highlighting the issue. Now corrected.

Line 180: The resampling and NDWI steps appear to be described in Figure 2 in the opposite order?

That was incorrect in text and correct on Figure 2. We've altered the text.

Line 231: Not an issue - I'm mostly just curious - why you chose NN here (and bilinear elsewhere)?

This was to avoid any smoothing of the values associated with bilinear interpolation, which we think would introduce errors into the depth and thus volume calculations.

Line 241: Well done with these acronyms

Thank you!

Figure 3 & 4: Include legend so figure is easier to read

**Done.**

Line 336: Are these symmetrical distributions? Perhaps using Quartile1 / Quartile3 info will help describe the data while also being sure to use non-parametric statistics.

This information has been added alongside a correction to the values presented in the original manuscript (which stemmed from including values of 0 in the original analysis).

Line 341: Did you ever compare with MODIS to make that the \*it\* failed the test (as it would be expected to)?

Unfortunately, we didn't do any direct MODIS comparisons in this study because we didn't have the 2016 MODIS imagery downloaded and processed (and we think the paper is already sufficiently detailed without any such comparisons), but it is certainly something that could be done in future work.

Figure 6/7: I really like how you can display the data in the context of which data is available!! Have you considered if there was any variability in lake distribution / more nuanced than just scaling by cloud cover percent? Like an elevation-dependent extrapolation, or something like that? Also: consider combining into one figure

Thanks for the suggestion to scale the lake data in other ways. We didn't do anything of this sort in this study, but it would certainly be possible to do, even though it would of course have an associated caveat that it would not necessarily result in an accurate representation of the data (as indeed is also true when scaling by cloud cover and regional data cover). To address the second part of the comment, we think it is preferable to keep these as two separate figures as they show two different sets of data that may otherwise be confusing, so have not combined them into a single figure.

Figures 8 & 9: Combine into one figure

Again, we think that these are better as two separate figures as they display different things altogether, so have not made this change.

Fig 11: These colors are not necessarily all distinguishable to a color-blind reader. Revise colors and/or label the lines on the righthand axis.

**Thanks for pointing that out. We have labelled the lines on the right-hand axis as per your suggestion.**

Line 578: What about sharing your resultant lake dataset? Line 578: The Cryosphere data policy encourages a few things that are different from how this paper handles sharing – most importantly share data AND code in an open, citable repository. Requiring to ask an author is a large barrier particularly in the future. See Gil et al. for further suggestions, and I'm sure that your readership would love to have these tools and dataset available in an open place / GitHub repo.

**https://agupubs.onlinelibrary.wiley.com/doi/full/10.1002/2015EA000136**

Thank you for this suggestion – we had overlooked the specific data policy of The Cryosphere. The full FASTER source code is now available publicly (and is appropriately referenced within the manuscript) with an embargo for a year to allow other pieces of ongoing work using this code to be completed. However, individuals are able to request the code via the link provided, and their requests will be approved by me. After the embargo expires, the full code will be available to anyone. Access to the source code allows other workers to reproduce the outputs of this work.

**Response to reviewer #2 (Samuel Doyle)**

Williamson et al. present a new technique which combines two different types of satellite imagery to create a lake geometry time series with an unprecedented combination of spatial and temporal resolution. The methods are rigorous and described in comprehensive detail. Indeed the detailed description of the methods gives the impression that the manuscript is largely a techniques paper despite significant results also being presented. The figures and tables are of a high standard and the referencing is appropriate throughout. The manuscript reads well with very few errors. The finding that a large proportion of surface water drains through small (and often previously undetected) lakes is significant. The paper builds on previous work on supraglacial lakes on the Greenland ice sheet.

We are very appreciative of the positive remarks, thank you.

**General comments**

My main concern stems from L213: "We treated these Landsat depths and volumes as ground-truth data as in Williamson et al. (2018)". This assumption is not discussed in detail at any point in the manuscript, although I guess it may have been described in the author's previous paper. In my opinion, there should be more discussion in this manuscript of the lack of actual ground truthing and how this might affect the absolute accuracy of the results. That is, how do the authors expect the results to compare with reality? How does this study compare to previous studies which employed ground truthing? To be clear, I'm not expecting ground truthing to be undertaken, but the lack of ground truthing and its potential effects should be discussed.

This was indeed discussed in one of my previous papers; however, we have now added an additional paragraph at the end of discussion Sect. 4.1 to include further justification for using the Landsat 8 data as ground-truth data, and the limitations that are inherent in their use as such. We also mention how future work may take this initial work forward, which would include comparing the Sentinel-2 lake-depth measurements with 'better' or other ground-truth data, such as higher-resolution satellite-imagery measurements or field data of lake depth. We hope that this addresses your concern sufficiently.

There is also a lack of discussion of the depth limitation of the techniques used. The techniques presented only retrieve lake depth up until a certain point. Figures 3 and 4 suggest a 6 m limit, when lakes are known to be often deeper than 10 m. If many lakes are deeper than the limit then this would hypothetically create a bias in the results by underestimating the volumes of the deeper lakes, which could affect the reliability of the main conclusion of the paper — that the water drained through small lakes is greater than that through large lakes. Given this, the relatively low maximum lake volumes ( $1.2 \times 10^7 \text{ m}^3$ ) reported on L334/335 compared to those in the literature (e.g. Box and Ski, 2007) are potentially concerning. These limitations should be discussed to make the reader aware of them. The authors could also look to previous studies, which have measured lake depths to understand whether the maximum depth of their technique is potentially too shallow.

Thanks for pointing this out. We agree that the depth and volume measurements are lower than those that have been observed in the field and this is likely to be due to the sensitivity of the physically based methods for measuring lake depth (i.e. the red wavelengths become fully attenuated in the water column). This effect is present for all remote-sensing imagery as far as we are aware, with previous studies showing the same for WorldView-2 (Moussavi et al., 2016), Landsat 8 (Pope et al., 2016) and MODIS

(Williamson et al., 2017). The same problem would not necessarily exist with empirical techniques for calculating lake depth remotely, e.g. a method to scale reflectance to depth based on field measurements of depth (e.g. Fitzpatrick et al., 2014); however, they require field data, and site- and time-specific tuning and are not physically based. We think that the limitations of the physically based technique for calculating lake depth are already sufficiently discussed in Sect. 4.1. But, we have now included an extra clause where we explain that the effect of these limitations may be an under-measurement of lake volume compared with previous work (e.g. Box and Ski, 2007). However, it should be noted that, while the maximum volumes are larger in their study, the lake volumes calculated using our technique are of the same order of magnitude as many of the other values presented in their study (e.g. their Table 5). We also think that the conclusions relating to the water drained after small lakes have opened-up moulins are independent of the method for calculating lake depth and volume (since they are based on the identification of lake drainage and the runoff data derived from a different data product), so overall do not think that this creates concern over the reliability of this conclusion.

These issues aside, the discussion of problems and limitations in Section 4.1 is well- considered and appropriate.

**Thank you.**

**Specific comments**

L24/25 — consider offering an explanation for why small lakes drain such a large proportion of the runoff. Is it because they are more numerous? An explanation for this observation is never given in the manuscript.

Clarified that this is indeed a product of the more numerous small lake-drainage events that allow more moulins to open and the fact that small lakes are usually at lower elevations.

L29 — the last two references in this sentence are in the wrong place. They currently only relate to there being two main ways the ice sheet loses mass (without specifying what they are), which as a statement does not need a citation. Perhaps move them to later in the sentence/paragraph.

**Moved to later in the paragraph.**

L40 — I'm not sure whether the citations to Doyle et al. (2018) and Hofstede et al. (2018) are appropriate here. Presumably they are given here as evidence for subglacial sediment? In fact the number of citations given here could be significantly reduced to list only the key studies.

Apologies, the referencing here was unclear: some of the references related to observations of enhanced sliding over hours to day associated with surface melt delivery to the bed, and some of the references related to the potential for increased flow speeds when meltwater reaches the bed if the Greenland Ice Sheet is underlain by sediment. This is now hopefully clearer in the revised manuscript.

L43 — Consider the recent paper by Christoffersen et al. (2018) here.

Included.

L44 — It is well established that surface water delivered to the bed of the Greenland ice sheet accelerates ice flow in the short term. Given the evidence for this, the phrase 'potentially explaining' seems a bit weak.

**Removed 'potentially'.**

L45 — Consider citing Joughin et al. (2013) and Hoffmann et al. (2011) here.

**Added.**

L75 — I'm not aware of any evidence which suggests lake hydrofracture takes days. All the evidence suggests the process is rapid, taking hours or less.

**Changed.**

L111 — May to October is not summer, perhaps use 'melt season' instead, although there is not usually much melt in October.

**Changed.**

L124 — Why does the study only cover up to 90 km inland? Perhaps give a reason here.

**Clarified in text. It's related to the size of the original Sentinel-2 tiles.**

L187 — stating the increments provides no information on how much it was adjusted in absolute terms, unless only one increment was used? Or is the percentage change given per 0.01 increment? Please clarify.

**Clarified in text.**

L230 — Can you (briefly) be a bit more specific here as to how the new technique was validated, even if it requires some repetition.

This information is included at the end of this paragraph in the sentence beginning: "To evaluate the performance ..." We have now also added an extra clause here to remind the reader of the validation procedure.

L260 — Consider discussing the implications of Cooley and Christoffersen (2017) regarding the effects of observation bias on the detection of rapidly draining lakes. The reduced interval of the new technique presented in the manuscript under review should reduce the observation bias associated with the longer intervals of previous studies.

**Thanks for that suggestion. We think that this comment fits better towards the end of discussion Sect. 4.3, so have added it there.**

L271 and L418 and L496 and elsewhere — The term 'interior' is here used as a synonym for 'englacial'. This is arguably ambiguous with the frequent use of 'interior' to describe the central region of the ice sheet (including the surface) away from the margin. The use of this term here also neglects the important effects of surface water reaching the bed.

**Altered to 'internal hydrological system' here and throughout.**

L335 — These maximum volumes seem quite low. How do they compare with the literature? Are there any reasons why the volumes are low? Is it due to the depth limitation of the techniques used?

This has now been discussed in Sect. 4.1. And please also see the response to the earlier comment relating to this.

L411 — The two modelling studies cited here were not the first to suggest this. Consider citing other studies and/or adding an e.g. before the citation.

**Added 'e.g.' before.**

L418 — Why is this? Why does more water drain through small lakes than larger ones? Can it be explained by the greater number of small lakes, or is it a result of drainage basin size, or is it a result of a bias in the technique? Some discussion is warranted.

This has now been clarified in the text (discussion Sect. 4.3) to be a result of the greater number of small lakes and the fact that they are found at lower elevations than large lakes, where melting is higher.

L503 and other occurrences of this pair of citations — these modelling studies may not be considered as the first or most appropriate for the establishment of drainage through moulins, which has been known for a long time (and was not determined by modelling). Consider listing earlier citations and/or giving an e.g. first to show that these citations are selected examples. Some of the sentences preceding these citations (including that on L503) may not even need a citation.

This has been clarified with an "e.g." has been added. We think that the citations in the previous line are required, so have not made a change.

L511 — 'runoff' not 'meltwater'.

Changed.

L512 — 'the moulins'.

Changed.

Fig. 1 — Consider labelling some glaciers to aid the reader. Also, consider showing enlarged images of each of the two example lakes as subplots to demonstrate the capability and resolution of the imagery.

Glaciers now labelled, thank you for suggesting. Enlarged image shown of the rapidly draining lake in the green box.

Fig. 3 — at this scale, whether the markers are circles or squares is redundant.

Changed reference to 'circles' and 'squares' to 'markers' where relevant, and now included a legend on the figure (as per reviewer #1's suggestion) to show this information.

Fig. 10 — consider a red/blue transparency with purple overlap (or any other primary colour pair).

Changed.

**Technical corrections**

L20 — 'identify' is the wrong word here, suggest 'estimate'

Changed.

L25 — strictly speaking its not via all moulins but only those identified within 'small' and 'large' lakes.

Changed.

L95 — define MSI.

Done.

L112 — the last sentence of this aim isn't written as an aim and there is change of tense from the adjacent sentences.

Changed.

L114 — define 'rapidly' here.

Done.

L208 and other occurrences - within the text write out 'Section' or 'Figure' in full.

This is contradictory to the journal's house standards (where Sect. and Fig. are used in running text, unless at the start of a sentence), so this change has not been made.

L258 — citation should be to Doyle et al. (2013).

Changed.

L297 — replace 'more poorly' with 'worse'.

Changed.

L330 — what does the number in brackets refer to?

Clarified.

L376 — Suggest: 'when the pair of images were only separated by a day'.

Changed.

L391 — "Large lakes are *defined as* . . ."

Changed.

L396 and L405 and L428 - Write out 'Figure' within text.

This is contradictory to the journal's house standards (where Sect. and Fig. are used in running text, unless at the start of a sentence), so this change has not been made.

L434 — consider rearranging to avoid double brackets.

Done.

L465 — 'entirely' is not necessary here.

**Removed.**

L490 — 'offset' is the wrong word here.

**Changed.**

Table 2 — Consistency with precision. Specifically, minimum drainage volume should be given as 0.020 for large lakes for Sentinel 2 (not 0.02). The same applies to the same for Landsat 8.

Changed.

Section S1 — delete 'the value of' in the first sentence.

Done.

Table S2 — Write out 'The asterisk denotes . . . '

Done.

Fig. S1 — Overlap in X-axis label superscript. Also, consider inserting 'therefore' in '. . . between the two sets of lake areas is *therefore* remarkably small'.

**Changed.**

**Additional references (not already cited in the manuscript)**

Christoffersen, P., Bougamont, M, Hubbard, A., Doyle, S.H., Grigsby, S. & Pettersson, P. 2018. Cascading lake drainage on the Greenland ice sheet triggered by tensile shock and fracture, *Nature Comms.*, 9 1064.

Cooley, S. W., & Christoffersen, P. (2017). Observation bias correction reveals more rapidly draining lakes on the Greenland Ice Sheet. *Journal of Geophysical Research: Earth Surface*, 122, 1867–1881

Joughin, I., Das, S., Flowers, G., Behn, M., Alley, R., King, M., Smith, B., Bamber, J., van den Broeke, M. & van Angelen, J. 2013. Influence of ice-sheet geometry and supraglacial lakes on seasonal ice-flow variability, *The Cryosphere*, 7, 1185-1192.

**Response to reviewer #3 (Kristin Poinar)**

**Summary**

Williamson et al describe a substantial new contribution to the remote-sensing detection of Greenland supraglacial lake drainage events. Their approach is to combine images from two medium- to high-resolution sensors to achieve near-daily time resolution of a study area in WNW Greenland. The authors apply this new technique over the 2016 melt season and are able to detect smaller draining lakes ( $

What, then, can explain the high (61.5%) contribution of the small lakes? Is it simply a large total size of their basins? This information would be easy to include (I believe it is already calculated).

This has now been clarified to be a result of both the lower elevation of small lakes compared with large ones, and the fact that there are more small lakes. In text, the mean and standard deviations of small lakes have been added (Sect. 4.3) to demonstrate this point. In addition, the slight difference in drainage dates likely has some effect, but we agree that this is likely to be only minimal, so we have not removed it entirely, but toned it down.

Finally, and most crucially, the conclusion that small lakes are important to the subglacial hydrological system is based on the assumption that their moulins stay open for the entire melt season (Figure 11). I don't have any especial reason to doubt this, but the assumption is not backed up in the paper although I believe the authors' data could easily do so. Presumably, if a moulin were to close up before the end of the melt season, a lake would re-form on site, and could be seen in the data. I think I can infer from the description of the FASTER algorithm that any such lakes would not meet the criteria for "fast-draining" and thus would not be included in this study – but this is not stated/explained in the manuscript.

We have now acknowledged the assumption that the moulins must remain open more clearly at several places in the manuscript, and in the discussion have stated that this may vary across the study region according to ice thickness or stress state, for example. The FASTER algorithm would filter out some lakes in the way you suggest since it filters lakes that re-fill on the subsequent day of cloud-free imagery by more than 20% of the total water volume lost during the drainage event. So, these lakes would not be included in the analysis. However, lakes that refill later in the season are not excluded from the "fast-draining" category in the way suggested above, so it is difficult to verify the assumption using the data available to us. In addition, testing this assumption using the FASTER algorithm in the way suggested assumes that there is sufficient melt to fill a basin after it has drained and the moulin has closed up. We think that acknowledging the assumptions inherent in our method is sufficient to justify the conclusion relating to the importance of small lakes.

**Technical comments**

Line 7 - I wouldn't start the abstract with "Although"; move it to the middle of the sentence as "however".

Changed.

Lines 47, 49 - It is not sensible heat, but latent heat that makes both of these effects (Phillips and Mankoff references). Water temperature is not important.

Removed references to temperature.

Line 49 - It's actually Poinar et al. 2016 (not 2017)

Apologies, but we think that our original reference was correct. We believe that the final publication date of the article in question was 2017, even though it was initially accepted and appeared online on the Journal of Glaciology website under FirstView in 2016. https://doi.org/10.1017/jog.2016.103 for details.

Line 67 - Replace "this" with its antecedent (since it is the first sentence of the paragraph).

Done.

Line 82 - The records have no problems; instead, perhaps the methods have shortcomings.

Altered.

Lines 85 and 95 - Define or remove SAR, MSI acronyms

Both defined.

Line 135 - Clarify year 2016

Done.

Lines 155, 159 - I got a bit confused with the numbers here, since they are similar (38 + 39 = 77). Perhaps recast the sentences to use only the number 39. Also please state the year 2016 again here.

Done.

Line 190 -  $R^2 = 0.999$ , that's excellent, good for you!

**Thank you!**

Line 271 and elsewhere - "GrIS interior" confused me; to me it means inland or upstream regions, whereas you intend to say that the water leaves the surface and enters the englacial or subglacial environment.

Clarified as 'internal hydrological system' throughout.

Line 294 - Here you say  $\sim$ 3 meters but later (line 465) you say  $\sim$ 3.5 meters.

Changed to 3.5 metres in both instances.

Lines 347-350 - This sentence would benefit from parallel construction.

Changed to add a colon to link the sentences to show that the second clause logically follows from the first.

Line 353 - p=0.00 should be more precise.

**Clarified, although the value we have now included is essentially meaningless as it is so small!**

Line 443 - "We opted for" sounds a bit informal.

Changed.

Lines 458-468 - Effect of July 1 "cloud adjacency". Pixels 200 m from clouds were already removed (Data and methods section), so it seems to me that this cloud-adjacency argument would not apply. Perhaps more description of how "adjacent" (i.e., if there are effects >200 m away) these effects are is required.

This has been clarified that there may have been effects at > 200 m distance from the clouds.

Lines 536-539 - It reads a little harsh on Miles et al. to end the paragraph with the shortcomings of that study; instead wouldn't it be better to end by highlighting the strengths of your own study?

**This paragraph has been restructured and reworded to deal with this comment.**

Lines 540-541 - As written, this sentence is false because other studies have combined two optical satellite datasets (e.g. MODIS and L8). Recast by adding "medium- resolution" or moving the sensors out of parentheses.

Changed.

Line 561 - "less" instead of "not"

**Changed.**

Figure 7. I like that Figure 6 was scaled up to the full image region here. The data show a lot of variability (sawtoothlike) on multi-day scales. I'd be interested in whether this is "real" (perhaps regionally linked lake drainages?) or just noise remaining from the effects of clouded-over regions.

This is essentially noise in the data. It likely stems from the fact that even if there is lower regional cover on a single day (e.g. large areas of cloud or no-data values), and the data are scaled accordingly with these values, this method does not take account of precisely where the missing data are relative to the locations of actual observations on the image. Of course, it is indeed likely that some of the changes are indeed volume loss from the surface due to lake drainage, but these are not separated out here. So, Figure 7 really just presents an estimate of how the data might look, with some of the patterns being unrealistic.

Figure 10. The magnitude of lake volume (x axis) seems much too large: the largest lake would have a volume of  $10^{16}$  m3, which would be 100 km x 100 km x 10 km, way too big. There must be an error here.

Apologies, this was an incorrectly labelled x-axis; these values were calculated by taking the natural logarithm of the original values, not the logarithm to the base 10. The label has been updated.

Figure 11. The y axis label is confusing: volume, yet mm?

This was incorrect, we apologise, and was only something we became aware of after the manuscript had entered review. We have now included an updated figure.

Table 2. Consider adding the mean surface elevation of the 3 classes of lakes, as described in the "Specific comments" section of this review.

The mean and standard deviation of the small and large lakes are now clarified in text.

[revised manuscript text omitted]

---

## Author Response (AR2)

Dear Bert,

Thank you for your comments on our manuscript tc-2018-56.

Below, we have responded to each of your comments sequentially. Here, your original comments are in normal font and our responses are in *italicised* font. In addition to the changes to the manuscript and supplement resulting from your comments, we have made a small number of additional changes during our final read-through of the documents; we hope that these are acceptable. All of the changes are shown on the tracked-changes versions of the manuscript and supplement below.

We look forward to receiving a final decision on the manuscript shortly.

Kind regards,

Andrew Williamson, on behalf of all authors
* * *
**Response to editorial comments**

line 103-105: Sentinel-2 data have been used for other glaciological applications as well, for example glacier flow velocity, identifying calving events, etc. Remove 'so far' and add 'and other applications' or similar at end of sentence.

*Thanks for pointing out – altered to note that these are only sample applications.*

line 202: 'in increments of ± 0.01 from these values' seems to be redundant since you are only using one increment positive and negative.

*Changed to indicate that it was only one increment in the positive and negative directions.*

line 246-249: long sentence. I suggest to change to "Our empirically based approach involved deriving various lake depth-reflectance regression relationships, to determine which explained most variance in the data. We used the Landsat 8 lake depth data (dependent variable) and the Sentinel-2 TOA reflectance data for the three optical bands (independent variables) for each pixel within the lake outlines predicted in both sets of imagery, to determine which band and relationship produced the best match between the two datasets."

*Changed in line with your suggestion, thank you.*

line 304: change "The physically based method (Figs. 3 and S2) applied to the red and green bands performed..." to "The physically based method applied to the red and green bands (Figs. 3 and S2, respectively) performed..."

*Changed.*

Fig 3, 4 and S2: Use a darker shade of blue/red for the 1 July/31 July legend. Also, include this legend in figures S3 and S4 for consistency.

*Done.*

line 410: "Additional...than" sounds a bit odd. Replace by "more...than"?

*Changed.*

line 459 and Fig. 11: "w.e." seems redundant as the (modelled) runoff is essentially liquid water

*True – now removed.*

line 465: change to "these SMALL lakes" for clarity

*Changed.*

line 589: would it be possible to calculate and include the % of rapidly draining large lakes only (i.e. only those that could be identified by MODIS)? This would provide a more fair comparison to Williamson 2017

*Added, thank you for the suggestion.*

[revised manuscript text omitted]